# LEARNAT: LEARNING NL2SQL WITH AST-GUIDED TASK DECOMPOSITION FOR LARGE LANGUAGE MODELS

**Weibin Liao**[1,2], **Xin Gao**[1,2], **Tianyu Jia**[1,2], **Rihong Qiu**[1,2], **Yifan Zhu**[6],
**Yang Lin**[7], **Xinyu Ma**[8], **Junfeng Zhao**[1,2,3], **Yasha Wang**[2,4,5*]

[1]School of Computer Science, Peking University
[2]Key Laboratory of High Confidence Software Technologies, Ministry of Education
[3]Big Data Technology Research Center, Nanhu Laboratory
[4]National Engineering Research Center For Software Engineering, Peking University
[5]Peking University Information Technology Institute (Tianjin Binhai)
[6]School of Computer Sciences, Beijing University of Posts and Telecommunications
[7]Huawei Technologies Co., Ltd
[8]Seed, ByteDance Inc.
✉ liaoweibin@stu.pku.edu.cn, wangyasha@pku.edu.cn
 https://github.com/MrBlankness/LearNAT.git

## ABSTRACT

Natural Language to SQL (NL2SQL) aims to translate natural language queries into executable SQL statements, offering non-expert users intuitive access to databases. While recent approaches leveraging large-scale private LLMs such as GPT-4 have achieved state-of-the-art results, they face two critical challenges: the lack of openness and reproducibility, and the prohibitive computational cost of test-time scaling. To address these issues, we explore improving the model-level performance of small-scale public LLMs in NL2SQL under resource-constrained settings. Our exploratory experiments reveal the potential of task decomposition for enhancing NL2SQL performance, but also highlight the difficulty of enabling LLMs to decompose queries effectively. Motivated by these findings, we propose `LearNAT`, a novel framework designed to enhance LLMs' decomposition capabilities. `LearNAT` introduces (1) a Decomposition Synthesis Procedure, which leverages AST-guided search with pruning strategies to generate verifiable and efficient decompositions, and (2) Margin-Aware Reinforcement Learning, which provides fine-grained preference optimization for multi-step reasoning beyond standard DPO. Extensive experiments on benchmark datasets demonstrate that `LearNAT` significantly improves the performance of small-scale LLMs, achieving results comparable to GPT-4 with only a 7B parameter model. These results validate the effectiveness of verifiable decomposition and fine-grained preference learning in advancing NL2SQL towards openness, transparency, and efficiency.

## 1 INTRODUCTION

Natural Language to SQL (NL2SQL) (Kim et al., 2020) is a fundamental task that seeks to automatically translate natural language queries into executable SQL statements (Zhang et al., 2024; Huang et al., 2025). This task has garnered substantial research interest owing to its potential to democratize database access, thereby enabling users without SQL expertise to query and interact with databases through natural language. In recent years, large language models (LLMs) (Lin et al., 2025), such as OpenAI's GPT-4, have achieved state-of-the-art performance on widely adopted NL2SQL benchmarks, including Spider (Yu et al., 2018) and BIRD (Li et al., 2023b). These approaches predominantly rely on large-scale proprietary LLMs, such as GPT-4 (Talaei et al., 2024; Lee et al., 2025;

---

*Corresponding Author.

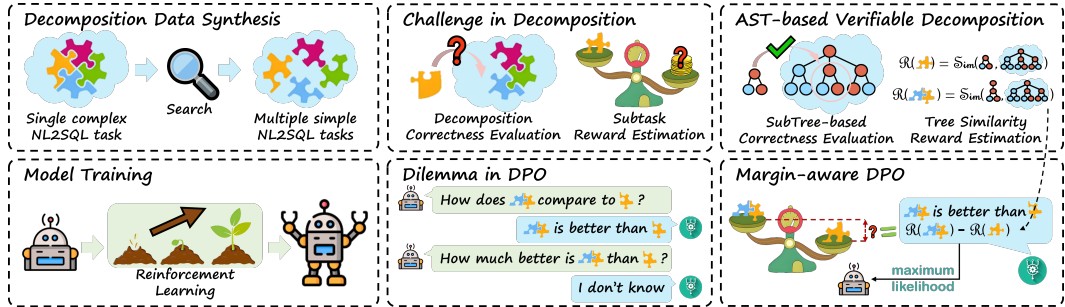

Figure 1: Overview of `LearNAT`. To address the challenges in decomposition data synthesis, `LearNAT` introduces a *Decomposition Synthesis Procedure* that enables AST-based verifiable decomposition. Furthermore, to overcome the limitation of DPO in capturing differences among subtask preferences during training, `LearNAT` proposes a *Margin-Aware Reinforcement Learning*, which leverages subtask-level rewards generated during the decomposition process to facilitate fine-grained and dynamic preference learning.

Wang et al., 2025) and Gemini (Pourreza et al., 2024), and often employ sophisticated prompt engineering techniques (Wei et al., 2022). Moreover, they leverage the test-time scaling law of LLMs to generate **system-level** outputs (as defined in Appendix G.1). Despite their effectiveness, this line of research encounters two critical challenges. First, the dependence on proprietary LLMs raises pressing concerns regarding openness, reproducibility, and data privacy. Second, the application of the test-time scaling law substantially increases computational overhead, which is particularly prohibitive in the context of large-scale LLMs. Consequently, this study emphasizes *enhancing the model-level performance (as defined in Appendix G.1) of small-scale public LLMs, with the overarching goal of fostering greater openness, transparency, and efficiency in NL2SQL under resource-constrained deployment scenarios.*

To improve the performance of small-scale public LLMs, prior research has explored strategies such as pre-training (Li et al., 2024b) and post-training (Yang et al., 2024b) to equip LLMs with domain-specific knowledge. However, NL2SQL tasks pose unique challenges. Natural language queries frequently encompass multiple objectives, which may be explicit (directly corresponding to query results) or implicit (e.g., conditions for filtering results), and these objectives are not always directly aligned with the underlying database schema. Such characteristics render it particularly **difficult for LLMs to effectively address complex NL2SQL tasks in a single step**. A promising direction, therefore, lies in decomposing complex NL2SQL problems into a sequence of simpler subproblems, thereby alleviating overall solution complexity.

To preliminarily validate this hypothesis, we conducted extensive exploratory experiments. These experiments yielded two key observations: (1) When subtasks are manually provided to the LLM, performance improves substantially (30.4%↑), underscoring the **potential of task decomposition** in enhancing NL2SQL performance. In contrast, when the LLM is responsible for decomposing complex queries on its own, the performance gains are marginal (3.4%↑), highlighting the need to **strengthen the task decomposition capabilities** of LLMs for NL2SQL. (2) We further investigated two feasible decomposition strategies: (i) AST-based decomposition with semantic verification and (ii) search-based decomposition with AST verification. Experimental results reveal that while the AST-based approach is computationally more efficient, it frequently introduces errors in the generated subtasks and fails to achieve precise semantic validation. Conversely, the search-based approach, though more complex, leverages AST-based verification to yield more stable and accurate task decomposition and translation. Collectively, these findings highlight **the critical importance of verifiable decomposition**.

Motivated by the above insights, we propose `LearNAT` (Learning NL2SQL with AST-guided Task Decomposition), a framework designed to enhance the decomposition capability of LLMs for complex NL2SQL tasks. It introduces two core components:

- *Decomposition Synthesis Procedure*: This verifiable decomposition component employs a search-based strategy, such as Monte Carlo Tree Search (MCTS), to generate subtasks for NL2SQL decomposition. Existing LLM-MCTS hybrid methods typically rely on heuristic evaluation, where

the LLM itself estimates node rewards to guide the search. However, even advanced models such as GPT-4 achieve only 46.35% accuracy on benchmarks like BIRD, limiting the reliability of such self-evaluation strategies. Moreover, the vast search space inherent in text-based MCTS introduces inefficiencies and computational overhead. To mitigate these challenges, we leverage abstract syntax trees (ASTs) to guide the search and implement pruning strategies, thereby substantially improving both efficiency and the success rate of generating valid decompositions.

- *Margin-Aware Reinforcement Learning*: This component enhances LLMs' decomposition capabilities through reinforcement learning techniques, such as Direct Preference Optimization (DPO) (Rafailov et al., 2023). Standard DPO algorithms struggle with fine-grained supervision in multi-step reasoning tasks, as they treat all positive and negative steps equally. To overcome this limitation, we propose an AST-based margin-aware DPO framework that differentiates between varying levels of step correctness, enabling more precise optimization.

Our main contributions can be summarized as follows:

1. **Conceptually**, we tackle the critical challenge of enabling LLMs to comprehend users' high-level semantics and map them to database schemas for complex NL2SQL queries. To this end, we propose LearNAT, the first framework to improve LLM performance on NL2SQL tasks by explicitly leveraging task decomposition.
2. **Methodologically**, we introduce the *Decomposition Synthesis Procedure*, which provides a verifiable and efficient decomposition mechanism that assesses both subtask correctness and overall task progress via ASTs, and *Margin-Aware Reinforcement Learning*, which enables fine-grained preference learning tailored to multi-step reasoning.
3. **Empirically**, through extensive experiments on two NL2SQL benchmark datasets, we demonstrate that LearNAT substantially outperforms existing approaches, achieving performance on par with GPT-4 while using a 7B-parameter model. These results underscore the effectiveness of task decomposition strategies in addressing the inherent challenges of complex NL2SQL tasks.

## 2 MOTIVATION FROM PRELIMINARY EXPERIMENTS

**Motivation for Task Decomposition.** In this empirical study, we conduct a detailed investigation to examine whether task decomposition benefits LLMs in the NL2SQL task and further explore the bottlenecks of decomposition-based NL2SQL approaches. Specifically, we randomly selected 500 examples from the BIRD-train (Li et al., 2023b) set and evaluated Qwen2.5-coder-32B (Yang et al., 2024a) under three experimental settings: (1) directly prompting Qwen2.5-coder-32B to solve the NL2SQL tasks without decomposition; (2) manually decomposing complex NL2SQL tasks into simpler subtasks and then prompting Qwen2.5-coder-32B to solve them; (3) using a prompting-based method to guide Qwen2.5-coder-32B to first decompose complex NL2SQL tasks into simpler subtasks and then solve these subtasks sequentially. The experimental results are provided in Appendix Fig. 6. The results reveal a striking contrast: when LLMs are guided with manually crafted subtask decompositions, their performance improves significantly—for instance, by 30.4%↑. However, when relying on LLMs to autonomously decompose tasks, the improvement is modest, with gains of only around 3.4%↑.

We posit the following: complex NL2SQL tasks can be effectively decoupled into two distinct subproblems: ***high-level task decomposition*** and ***low-level NL2SQL translation***. The former involves breaking down a complex user query into a sequence of simpler, manageable subtasks, a process that often demands substantial reasoning capabilities. The latter focuses on directly translating these simplified natural language inputs into their corresponding SQL queries. Owing to extensive pre-training, LLMs generally excel at low-level NL2SQL translation. However, they are far less proficient at complex task decomposition, as they often lack the deep reasoning and planning abilities required to deconstruct intricate problems into logical steps. Based on these findings, we derive our first key observation:

***Observation I:*** *Task decomposition holds substantial promise for enhancing the NL2SQL capabilities of LLMs. Nevertheless, the limited ability of LLMs to perform effective task decomposition autonomously constitutes a major bottleneck to the practical deployment of decomposition-based NL2SQL techniques.*

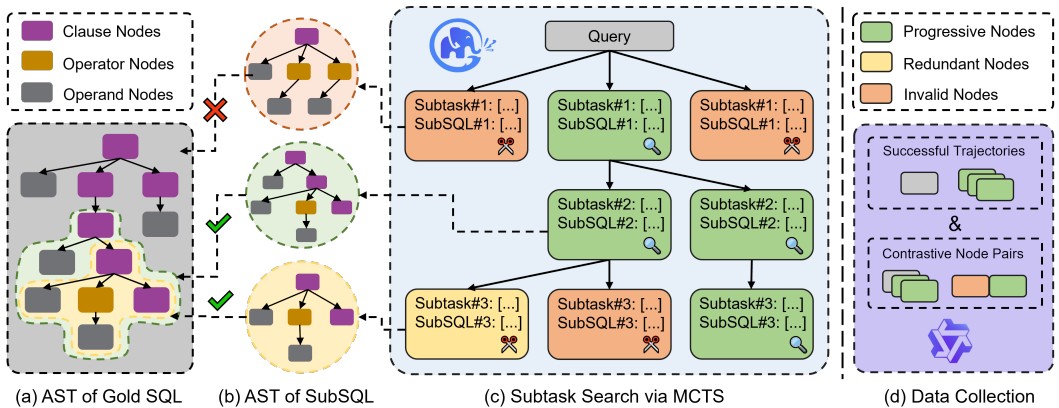

(a) AST of Gold SQL    (b) AST of SubSQL    (c) Subtask Search via MCTS    (d) Data Collection

Figure 2: Framework of the *Decomposition Synthesis Procedure*. (c) illustrates how the LLM, combined with MCTS, performs next-step prediction to synthesize subtasks of complex NL2SQL tasks. (b) presents the AST of the SQL statements corresponding to each synthesized subtask in (c). (a) shows the AST of the Gold SQL for the complex NL2SQL task, which guides the MCTS in (c) to perform more efficient search, including pruning and node reward estimation. (d) depicts the data collected by `LearNAT` during the *Decomposition Synthesis Procedure*, comprising successful trajectories data for supervised fine-tuning and step-wise contrastive node pairs data for preference learning. Under the default settings of `LearNAT`, GLM-4-Plus is used to synthesize decomposition data, and the Qwen2.5-Coder model is fine-tuned.

**Motivation for Verifiable Decomposition.** A critical challenge in leveraging task decomposition to enhance NL2SQL performance in LLMs lies in **how to derive a sequence of simpler subtasks from a complex NL2SQL query**. To address this issue, we explored a straightforward hypothetical solution in our preliminary experiments. Specifically, as illustrated in Appendix Fig. 7 (a), we decomposed the ground-truth SQL corresponding to a complex natural language query into several simpler sub-SQL queries based on its abstract syntax tree (AST). We then employed an LLM to translate these sub-SQLs into natural language subtasks, treating them as the subtask sequence for the original query. However, this approach introduces a new challenge: owing to the hallucination tendencies of LLMs, the translation process from sub-SQLs to subtasks may introduce latent errors. This raises a crucial question—**how can we verify the correctness of the generated subtasks?**

We attempted to directly use language models, such as Sentence-Transformer and GLM-4, to determine whether the generated subtasks were correct based on semantic similarity. However, these approaches achieved only 46.8% and 36.0% accuracy, respectively (see Appendix Table 5), indicating that verifying the correctness of subtasks is far from straightforward. The language models frequently misjudged cases involving subtle semantic differences. For example, if the gold SQL query targeted a user's name while the generated subtask query instead targeted the user's ID, GLM-4 often incorrectly deemed the subtask to be correct. This motivates our second key observation:

*Observation II: Subtasks generated by large language models are often unreliable, thereby necessitating a verifiable task decomposition procedure.*

**Summary.** Building on the above observations, our objective is to develop a verifiable task decomposition framework and leverage reinforcement learning to strengthen the decomposition capabilities of LLMs in NL2SQL. To this end, we propose `LearNAT`, which consists of two key components: (1) a *Decomposition Synthesis Procedure* that generates candidate subtask sequences via search-based decomposition and evaluates their correctness using AST-based validation (see Appendix Fig. 7 (b), addressing *Observation II*); and (2) a *Margin-Aware Reinforcement Learning* framework that enhances LLMs' task decomposition ability by incorporating step-level task awareness, thereby improving overall NL2SQL performance (addressing *Observation I*).

## 3 METHODOLOGY

In this section, we present the methodology of `LearNAT`. First, `LearNAT` employs the *Decomposition Synthesis Procedure* for generating training data in offline reinforcement learning. Then,

it utilizes *Margin-aware Reinforcement Learning* for model fine-tuning. For friendly reading, we provide preliminary knowledge and relevant notation tables for NL2SQL, AST, MCTS, and DPO in the Appendix. B.

### 3.1 DECOMPOSITION SYNTHESIS PROCEDURE

**Problem Formulation.** Let $\{q_1, q_2, \cdots, q_n\}$ denote a sequence of subtask queries, where $n$ represents the number of subtasks and each $q_i$ represents a natural language query that captures a component of the original query $Q$. For each subtask query $q_i$, *Decomposition Synthesis Procedure* generates a corresponding SQL query $y_i$. The objective is to find a sequence of subtask queries such that their corresponding SQL queries collectively construct the target SQL query $Y$.

**MCTS-based Decomposition.** *Decomposition Synthesis Procedure* formulates the decomposition process as a tree search problem, and performs next-step prediction as action $a$ in each state $s$. In the Monte Carlo Tree, the root node represents the original query $Q$, each non-root node represents the state of executing the next subtask, and each path from root node to a leaf node represents a decomposition sequence.

At each state in MCTS, the *Decomposition Synthesis Procedure* employs an LLM to generate the next subtask $q_i$ and sub-SQL $y_i$. Formally, each state $s_i = \{q_i, y_i, \mathcal{AT}(y_i), \mathcal{AT}_{\text{sum}}(y_i), \mathcal{R}(s_i)\}$, where $\mathcal{AT}(y_i)$ is the AST of $y_i$, $\mathcal{AT}_{\text{sum}}(y_i)$ is the merged AST summarizing all nodes from root to node $s_i$ in MCTS, and $\mathcal{R}(s_i)$ is the reward estimation of $s_i$. The $\mathcal{AT}_{\text{sum}}(y_i)$ is mathematically defined as follows:

$$\mathcal{AT}_{\text{sum}}(y_i) = (\mathcal{N}_{\text{sum}}, \mathcal{E}_{\text{sum}}), \tag{1}$$

$$\mathcal{N}_{\text{sum}} = \bigcup_{j=1}^{i} \mathcal{N}(\mathcal{AT}(y_j)), \mathcal{E}_{\text{sum}} = \bigcup_{j=1}^{i} \mathcal{E}(\mathcal{AT}(y_j)). \tag{2}$$

**Node Classification.** *Decomposition Synthesis Procedure* classifies nodes into three categories based on their AST properties for subsequent prune strategy:

- **Progressive Nodes**: Nodes where $\mathcal{AT}(y_i)$ is a subtree of $\mathcal{AT}(Y)$ and $\mathcal{AT}(y_i)$ is not a subtree of $\mathcal{AT}_{\text{sum}}(y_{\text{parent}(i)})$. These nodes contribute new information toward the target SQL. For two ASTs $\mathcal{AT}_1 = (\mathcal{N}_1, \mathcal{E}_1)$ and $\mathcal{AT}_2 = (\mathcal{N}_2, \mathcal{E}_2)$, We define the subtree relationship as follows:

$$\text{isSubtree}(\mathcal{AT}_1, \mathcal{AT}_2) = \begin{cases} 1, & \text{if } \mathcal{N}_1 \subseteq \mathcal{N}_2 \text{ and } \mathcal{E}_1 \subseteq \mathcal{E}_2 \\ 0, & \text{otherwise} \end{cases}. \tag{3}$$

- **Redundant Nodes**: Nodes where $\mathcal{AT}(y_i)$ is a subtree of $\mathcal{AT}(Y)$ but is also a subtree of $\mathcal{AT}_{\text{sum}}(y_{\text{parent}(i)})$. These nodes provide no additional reward to the decomposition.
- **Invalid Nodes**: Nodes where $\mathcal{AT}(y_i)$ is not a subtree of $\mathcal{AT}(Y)$. These nodes represent incorrect decompositions.

**Prune Strategy.** In traditional MCTS, since the typical scenario involves robotic task execution, $\mathcal{A}(s)$ is generally defined as a finite action set, such as `pick up`, `put down`, etc. However, in the application of LLMs, $\mathcal{A}(s)$ is usually an infinite action set. This is because LLMs generate actions in the form of text, meaning that even the same subSQL can be expressed as multiple different subtask (action) variations. To reduce the search space of MCTS and improve search efficiency, *Decomposition Synthesis Procedure* adopts a pruning strategy. Specifically, since the subtask sequence collected by the *Decomposition Synthesis Procedure* corresponds to the action sequence along the path from the root node to a leaf node in MCTS, redundant actions and invalid actions along the path do not need to be included in the subtask list. Therefore, for states containing redundant or invalid actions, the *Decomposition Synthesis Procedure* terminates further action searches to perform pruning.

**Reward Estimation.** In MCTS, it is necessary to estimate $Q(s, a)$ for each state to provide state rewards, thereby guiding the direction of subsequent searches. In general mathematical domains, existing works typically employ either LLM-based self-evaluation or an additional reward model trained for state reward estimation. In this work, the *Decomposition Synthesis Procedure* further leverages information from the AST and designs a rule-based approach to evaluate the state reward.

Since states with redundant actions and invalid actions are pruned, to improve efficiency, reward estimation is only performed for states with progressive actions. Specifically, *Decomposition Synthesis Procedure* estimates the reward of the current state based on the similarity between $\mathcal{AT}(Y)$ and $\mathcal{AT}_{\text{sum}}(y_i)$ at the current state.

$$\mathcal{R}(s_i) = \text{sim}(\mathcal{AT}_{\text{sum}}(y_i), \mathcal{AT}(Y)), \tag{4}$$

where $\text{sim}(\cdot, \cdot)$ denotes the AST similarity measure.

*Decomposition Synthesis Procedure* defines two types of AST similarity, including node-level similarity $\text{sim}_{\text{node}}$ and structural similarity $\text{sim}_{\text{struct}}$:

$$\mathcal{R}(s_i) = \alpha \cdot \text{sim}_{\text{node}}(\mathcal{AT}_{\text{sum}}(y_i), \mathcal{AT}(Y)) + (1 - \alpha) \cdot \text{sim}_{\text{struct}}(\mathcal{AT}_{\text{sum}}(y_i), \mathcal{AT}(Y)), \tag{5}$$

where $\alpha$ are adjustment factors for the two types of AST similarity. Node-level similarity is defined by the degree of node overlap, while structural similarity is measured using the tree edit distance, a detailed description is provided in Appendix. C.

**Self-improvement Demonstration.** To improve the success rate of decomposition, *Decomposition Synthesis Procedure* employs few-shot learning and adopts adaptive demonstrations from the previous round. Specifically, it constructs a demonstration pool, which consists of samples that were successfully decomposed in the previous $i - 1$ rounds. Given a new task decomposition query, the procedure computes the AST similarity between the query and each query in the demonstration pool. It then selects the top-3 most similar queries as demonstrations to be included in the prompt.

**Data Collection.** During the search process, *Decomposition Synthesis Procedure* collect two types of data for subsequent offline reinforcement learning:

- **Successful Trajectories**: Sequences of $\{(q_1, y_1), \cdots, (q_n, y_n)\}$ that successfully decompose the target SQL, used for supervised fine-tuning.
- **Contrastive Node Pairs**: Pairs of incorrect node $(q_i^l, y_i^l)$ and their corresponding correct node $(q_i^w, y_i^w)$, used for preference learning.

### 3.2 MARGIN-AWARE REINFORCEMENT LEARNING

`LearNAT` proposes a *Margin-aware Reinforcement Learning* framework to train the LLM for decomposing complex NL2SQL tasks into simpler subtasks. The framework consists of two phases. First, *Margin-aware Reinforcement Learning* fine-tunes the LLM in a supervised manner based on correct decomposition trajectories, enhancing the model's ability to perform task decomposition and generate the correct output format. Then, *Margin-aware Reinforcement Learning* conducts direct preference optimization (DPO) with AST margin on the LLM using contrastive node pairs, suppressing incorrect subtask outputs and achieving finer-grained preference alignment.

**Warm-up Strategy for Foundational Skill Acquisition.** Given the training data from *Decomposition Synthesis Procedure*, *Margin-aware Reinforcement Learning* first performs supervised fine-tuning on successful decomposition trajectories. In a training instance $(Q, \mathcal{DB}, \mathcal{K}, \{(q_1, y_1), \cdots, (q_n, y_n)\})$, *Decomposition Synthesis Procedure* treats $[Q, \mathcal{DB}, \mathcal{K}]$ as the prompt $x$ and $\{(q_1, y_1), \cdots, (q_n, y_n)\}$ as the target response $t$, so the supervised fine-tuning objective is to minimize the log-likelihood loss:

$$\mathcal{L}_{\text{SFT}} = -\mathbb{E}_{(\boldsymbol{x}, \boldsymbol{t})} \left[ \sum_{i=1}^{I} \log p_\theta \left( t_i \mid \boldsymbol{t}_{1:i-1}, \boldsymbol{x} \right) \right], \tag{6}$$

where $\theta$ represents the fine-tuned LLM parameters, and $p_\theta(\boldsymbol{t} \mid \boldsymbol{x}) = \prod_{i=1}^{I} p_\theta \left( t_i \mid \boldsymbol{t}_{<i}, \boldsymbol{x} \right)$ is the conditional probability distribution of target subtask & subSQL sequence $t$ given prompt $x$. $I$ is the sequence length of $t$, and $i$ is the auto-regressive decoding step.

**DPO with AST Margin.** A phenomenon of pessimism suggests that the positive feedback provided by SFT alone cannot prevent LLMs from generating erroneous reasoning pathways. Existing

Table 1: Results of *Decomposition Synthesis Procedure*. The decomposition success rate and token consumption on BIRD-train are reported.

| Methods | Success Rate | Token Cost |
|---|---|---|
| CoT | 59.07% | 16,735K |
| MCTS | 71.55% | 334,694K |
| + AST Guide | 78.01% | 133,877K |
| + Self-improvement Demonstration | | |
| (1 round) | 79.33% | 137,456K |
| (2 round) | 79.73% | 142,017K |
| (3 round) | 80.00% | 145,977K |

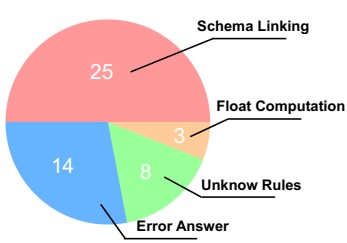

Figure 3: Error distributions of *Decomposition Synthesis Procedure* on randomly selected 50 error cases from the BIRD-train.

works (Rafailov et al., 2023) indicates that, during the SFT phase, as the probability of preferred outputs (correct responses) increases, the probability of dispreferred outputs (incorrect responses) rises as well. *Margin-aware Reinforcement Learning* employs DPO to suppress incorrect subtask outputs. Specifically, a training instance takes the form of $(Q, \mathcal{DB}, \mathcal{K}, \{(q_1, y_1), \cdots, (q_{i-1}, y_{i-1})\}, (q_i^w, y_i^w), (q_i^l, y_i^l))$, *Margin-aware Reinforcement Learning* treats $[Q, \mathcal{DB}, \mathcal{K}, \{(q_1, y_1), \cdots, (q_{i-1}, y_{i-1})\}]$ as the prompt $x$, $(q_i^w, y_i^w)$ as the preferred response, $(q_i^l, y_i^l)$ as the dispreferred response, and optimizes $\theta$ using DPO loss (see Eq. 13).

To enable finer-grained preference learning, *Margin-aware Reinforcement Learning* incorporates an offset into the DPO loss to measure the reward margin between positive and negative samples. The margin is directly computed using reward estimation based on AST similarity, eliminating the need for training an additional reward model. Specifically, *Margin-aware Reinforcement Learning* estimates the reward margin between two samples as follows:

$$\text{margin}((q_i^w, y_i^w), (q_i^l, y_i^l)) = \mathcal{R}(s_i^w) - \mathcal{R}(s_i^l). \tag{7}$$

Finally, the loss of DPO with AST Margin is formulated as follows:

$$\mathcal{L}_{\text{MDPO}}(\pi_\theta; \pi_{\text{ref}}) = -\mathbb{E}_{(x, y_w, y_l) \sim \mathcal{D}} \left[ \log \sigma \left( \hat{r}_\theta(x, y_w) - \hat{r}_\theta(x, y_l) - \triangle_r \right) \right], \tag{8}$$

where $\triangle_r = \text{margin}((q_i^w, y_i^w), (q_i^l, y_i^l))$ is the offset, measuring the reward margin between positive and negative samples. The AST margin effectively guides the model to learn not only which decomposition steps are preferred, but also how much they are preferred, leading to more nuanced and effective multi-step reasoning capabilities.

## 4 EXPERIMENTS

### 4.1 EXPERIMENTAL SETUP

**Datasets.** We use the BIRD-train dataset (Li et al., 2023b) to synthesize decomposition data for complex NL2SQL tasks within the *Decomposition Synthesis Procedure*, which is subsequently employed for *Margin-Aware Reinforcement Learning*. Then, we utilize BIRD-dev (Li et al., 2023b) (In-Domain) and Spider-dev (Yu et al., 2018) (Out-of-Domain) to evaluate the effectiveness and robustness of LearNAT. Notably, the databases and user questions in the training and test sets differ completely. A detailed introduction to the datasets and their statistical information is provided in Appendix. D.

**Evaluation Metrics.** Since the SQL expression styles generated by LLMs may differ from the ground truth in NL2SQL benchmarks (Shin et al., 2021), traditional string-based evaluation metrics, such as Exact Match Accuracy (Yu et al., 2018), are not suitable for our study. Therefore, following prior works (Liu et al., 2023; Rajkumar et al., 2022; Fan et al., 2024a), we adopt the Execution Accuracy (EX) metric, which evaluates the correctness of generated SQL queries by comparing their execution results with those of the corresponding ground-truth queries retrieved from the database. For additional experimental details, please refer to Appendix. F.1.

Table 2: Performance comparison on Spider-dev and BIRD-dev benchmarks. All baseline results are taken directly from the performance reported on Leaderboard. **Bold** indicates the best result, while underline denotes the second-best results achieved by `LearNAT`.

| Methods | Venue | LLMs | BIRD-dev (In-Domain) | | | | Spider-dev (Out-of-Domain) | | | | |
|---------|-------|------|--------|----------|-------------|-------|------|--------|------|-----------|-------|
| | | | Simple | Moderate | Challenging | Total | Easy | Medium | Hard | Extra Hard | Total |
| *System-Level* | | | | | | | | | | | |
| C3-SQL | | GPT-4 | 58.9 | 38.5 | 31.9 | 50.2 | 92.7 | 85.2 | 77.6 | 62.0 | 82.0 |
| DIN-SQL | NeurIPS'23 | GPT-4 | | | | 50.7 | 91.1 | 79.8 | 64.9 | 43.4 | 74.2 |
| MetaSQL | ICDE'24 | GPT-4 | | | | | 91.1 | 74.7 | 64.1 | 36.1 | 69.6 |
| MAG-SQL | | GPT-4 | 65.9 | 46.2 | 41.0 | 57.6 | | | | | 85.3 |
| SuperSQL | VLDB'24 | GPT-4 | 66.9 | 46.5 | 43.8 | 58.5 | 94.4 | 91.3 | 83.3 | 68.7 | 87.0 |
| MAC-SQL | COLING'25 | GPT-4 | 65.7 | 52.7 | 40.3 | 59.4 | | | | | 86.7 |
| *Model-Level* | | | | | | | | | | | |
| ACT-SQL | EMNLP'23 | GPT-4 | | | | | 91.1 | 79.4 | 67.8 | 44.0 | 74.5 |
| CatSQL | VLDB'23 | N/A | | | | | 95.8 | 88.3 | 74.7 | 62.7 | 83.7 |
| DAIL-SQL | VLDB'24 | GPT-4 | 62.5 | 43.2 | 37.5 | 54.3 | 90.3 | 81.8 | 66.1 | 50.6 | 76.2 |
| SENSE | ACL'24 | CodeLLaMA-13B | | | | 55.5 | 95.2 | 88.6 | 75.9 | 60.3 | 83.5 |
| CodeS | SIGMOD'24 | CodeS-7B | 64.6 | 46.9 | 40.3 | 57.0 | 94.8 | 91.0 | 75.3 | 66.9 | 85.4 |
| | | CodeS-15B | 65.8 | 48.8 | 42.4 | 58.5 | 95.6 | 90.4 | 78.2 | 61.4 | 84.9 |
| *Ours* | | | | | | | | | | | |
| `LearNAT` | | Qwen2.5-Coder-7B | 65.4 | 48.4 | 42.4 | 58.1 | 95.2 | **92.4** | 76.4 | 67.5 | 86.4 |
| | | Qwen2.5-Coder-14B | 68.5 | 51.4 | 45.8 | 61.2 | 95.6 | 91.5 | 80.5 | 68.7 | 86.9 |
| | | Qwen2.5-Coder-32B | **70.7** | **55.5** | **59.0** | **65.0** | **96.4** | **92.4** | **85.1** | **69.3** | **88.4** |

**Baselines.** In this experiment, we compare two types of baselines, including 8 system-level approaches and 7 model-level approaches. A detailed introduction to the baseline models and their statistical information is provided in Appendix. E. We consider two complementary evaluation strategies—*Competition on the Public Leaderboard* and *Comparison under Identical Evaluation Protocol*—to comprehensively assess the superior performance of `LearNAT`.

## 4.2 EXPERIMENTAL RESULTS

**Results of *Decomposition Synthesis Procedure*.** We evaluated the decomposition performance of the *Decomposition Synthesis Procedure* on BIRD-train and compared it with several baseline decomposition algorithms, including CoT and naive MCTS. The experimental results are shown in Table. 1. The results indicate that the *Decomposition Synthesis Procedure* achieved an 80.00% decomposition success rate, outperforming CoT and naive MCTS by 20.93%↑ and 8.45%↑, respectively. Additionally, it is noteworthy that MCTS generated a large number of invalid searches, leading to excessive token consumption. In contrast, our proposed *Decomposition Synthesis Procedure* utilizes AST-guided pruning, enabling high-performance and low-cost (56.38%↓) decomposition synthesis. We further tested the performance of self-improving demonstrations over multiple rounds. The results show that adaptive demonstrations significantly improve model performance (1.99%↑). However, this strategy also has inherent limitations. Table. 1 reveals that self-improving demonstrations achieved notable performance gains in the first round (1.32%↑), but in the subsequent two rounds, the decomposition performance began to diminish (only 0.4%↑ and 0.27%↑). Therefore, to minimize token consumption, we did not proceed with a fourth round of decomposition.

To further investigate the reasons for the failure of the *Decomposition Synthesis Procedure* in certain cases, we randomly selected 50 unsuccessful cases for error analysis. The error distribution is shown in Fig. 3. We analyze these errors one by one by presenting typical cases for each of the four error attributions in Appendix. F.2.

**Competition on the Public Leaderboard.** We evaluate `LearNAT` on Spider-dev[1] and BIRD-dev[2] benchmarks. To further assess `LearNAT`'s robustness, we fine-tune Qwen2.5-Coder models with 7B, 14B, and 32B parameters. Additionally, we compare `LearNAT` against recent competitive baselines from the past two years on leaderboard. The results are presented in Table. 2.

Compared with system-level methods, `LearNAT`—even with only a 7B model—already outperforms most approaches, although these approaches leverage larger-scale models such as GPT-3.5 or GPT-4 as backbone LLMs. For example, on the Spider-dev dataset, `LearNAT` (7B) achieves an overall accuracy of 86.4%, outperforming eight baseline methods and ranking just behind MAC-

---

[1] https://yale-lily.github.io/spider
[2] https://bird-bench.github.io/

Table 3: Performance comparison of `LearNAT` and competitive literature under identical evaluation protocol. **Bold** indicates the better result.

| Methods | Evaluation Protocol | LLMs | BIRD-dev (In-Domain) | | | | Spider-dev (Out-of-Domain) | | | | | Total |
|---------|--------------------|------|--------|----------|-------------|-------|------|--------|------|-----------|-------|-------|
| | | | Simple | Moderate | Challenging | Total | Easy | Medium | Hard | Extra Hard | Total | |
| SynCoT | SynCoT | Qwen2.5-7B-Instruct | | | | 59.2 | | | | | 78.9 | 67.1 |
| `LearNAT` | SynCoT | Qwen2.5-Coder-7B | 67.6 | 48.0 | 45.8 | **59.6** | 91.9 | 91.0 | 71.3 | 63.9 | **83.6** | **69.2** |
| OmniSQL | OmniSQL | Qwen2.5-7B-Instruct | | | | **63.9** | | | | | 81.2 | 70.9 |
| `LearNAT` | OmniSQL | Qwen2.5-7B-Instruct | 68.2 | 50.3 | 50.7 | 61.1 | 96.0 | 91.5 | 77.6 | 65.1 | **86.0** | **71.1** |
| SQL-o1 | SQL-o1 | Llama3-8B | 71.8 | 52.3 | 45.2 | 63.4 | 94.4 | 93.0 | 81.0 | 68.7 | 87.4 | 73.1 |
| `LearNAT` | SQL-o1 | Qwen2.5-Coder-7B | **72.5** | **54.2** | **49.3** | **64.8** | **96.4** | **94.8** | **78.2** | **74.1** | **89.1** | **74.6** |
| Alpha-SQL | Alpha-SQL | Qwen2.5-Coder-7B | 72.6 | 59.3 | 53.1 | 66.8 | 94.0 | 89.2 | 76.4 | 63.3 | 84.0 | 73.7 |
| `LearNAT` | Alpha-SQL | Qwen2.5-Coder-7B | **74.4** | **61.5** | 52.8 | **68.4** | **97.2** | **96.0** | **80.5** | **77.1** | **90.6** | **77.4** |

SQL and SuperSQL. The larger variant, `LearNAT` (32B), reaches 88.4% overall accuracy, surpassing all listed system-level approaches and outperforming the second-best method, SuperSQL, by 1.4%↑. Similar trends are observed on the BIRD-dev dataset. It is worth noting that although `LearNAT` (7B) performs slightly worse than MAC-SQL and SuperSQL, both of these belong to the system-level category. They involve more complex NL2SQL pipelines, including generating multiple candidates, SQL refinement, and consistency checks, which typically incur high token overhead. In contrast, `LearNAT` (7B), as a model-level method, generates the final SQL in a single forward pass without requiring additional token consumption. While maintaining the efficiency of model-level generation, it not only outperforms all other model-level baselines but also exceeds several system-level methods. This highlights `LearNAT`'s superior trade-off between performance and token cost.

Compared to model-level methods, `LearNAT` demonstrates a more significant performance advantage. Among the fine-tuning-based approaches mentioned, the most competitive is CodeS, therefore we evaluate both the 7B and 15B versions of CodeS. Experimental results show that `LearNAT` (7B) achieves a 1.0%↑ on Spider-dev and a 1.1%↑ on BIRD-dev over CodeS (7B). Similarly, `LearNAT` (14B) outperforms CodeS (15B) by a 2.0%↑ on Spider-dev and a 2.7%↑ on BIRD-dev. This indicates that `LearNAT` maintains a performance advantage across different model sizes.

**Comparison under Identical Evaluation Protocol.** We further compare the performance of `LearNAT` with a wider range of competitive prior works, including SynCoT (Liu et al., 2025), SQL-o1 (Lyu et al., 2025), Alpha-SQL (Li et al., 2025a), and OmniSQL (Li et al., 2025b). Because these baselines employ different evaluation protocols. For example, SynCoT triggers multi-step reasoning during inference, while SQL-o1 and Alpha-SQL perform MCTS-based search at inference time to generate additional candidate SQLs. We adopt the checkpoints of `LearNAT` and evaluate it under the same protocols as these baselines to ensure a fairer comparison. The experimental results are presented in Table. 3. The results show that `LearNAT` consistently surpasses SynCoT, SQL-o1, and Alpha-SQL, and although its performance on the BIRD dataset is slightly lower than OmniSQL, `LearNAT` achieves superior overall performance across both Spider and BIRD.

Notably, the overall performance of `LearNAT` can be further improved if additional search is introduced during inference. For example, when applying the inference-time search strategy of Alpha-SQL, `LearNAT`'s performance on the BIRD dataset increases from 58.1% to 68.4%. However, we also note a significant concern: the token consumption during inference rises sharply, from 1.8K tokens per query to 204.5k[3] tokens per query.

Both `LearNAT` and OmniSQL optimize NL2SQL performance by constructing synthetic datasets. However, OmniSQL focuses on synthesizing a much larger training dataset with 2.5M queries, whereas `LearNAT` emphasizes constructing training data with verifiable intermediate subtasks, improving data quality without increasing data quantity. Specifically, since `LearNAT` generates subtask data based on BIRD-Train, the resulting training set contains only 7.2k queries.

**Ablation Study.** We evaluate the necessity of each component in `LearNAT` by systematically removing individual components and assessing the model's performance. We use Qwen2.5-Coder-7B as the backbone LLM and conduct evaluations on Spider-dev and BIRD-dev. The results are summarized in Table. 4. Experimental results show that removing or replacing any single component

---

[3] https://openreview.net/forum?id=kGg1ndttmI

Table 4: Ablation study analysis of `LearNAT` using Qwen2.5-Coder-7B as backbone LLM. The green font indicates the performance loss incurred after the removal of the respective module.

| Methods | BIRD-dev (In-Domain) | | | | Spider-dev (Out-of-Domain) | | | | |
|---|---|---|---|---|---|---|---|---|---|
| | Simple | Moderate | Challenging | Total | Easy | Medium | Hard | Extra Hard | Total |
| `LearNAT` | 65.4 | 48.4 | 42.4 | 58.1 | 95.2 | 92.4 | 76.4 | 67.5 | 86.4 |
| *LearNAT* | | | | | | | | | |
| w/o `LearNAT` | 56.1 (9.3↓) | 34.5 (13.9↓) | 33.8 (8.6↓) | 47.5 (10.6↓) | 82.7 (12.5↓) | 84.1 (8.3↓) | 71.8 (4.6↓) | 54.8 (12.7↓) | 77.0 (9.4↓) |
| `LearNAT`→DPO | 61.7 (3.7↓) | 40.6 (7.7↓) | 34.7 (7.6↓) | 52.8 (5.3↓) | 84.7 (10.5↓) | 86.1 (6.3↓) | 74.1 (2.3↓) | 56.6 (10.8↓) | 79.0 (7.4↓) |
| `LearNAT`→CoT | 57.3 (8.1↓) | 37.6 (10.8↓) | 36.1 (6.3↓) | 49.3 (8.7↓) | 87.1 (8.1↓) | 85.2 (7.2↓) | 75.3 (1.1↓) | 56.6 (10.8↓) | 79.4 (7.0↓) |
| *Decomposition Synthesis Procedure* | | | | | | | | | |
| w/o AST Guide | 62.6 (2.8↓) | 41.5 (6.9↓) | 29.2 (13.2↓) | 53.1 (5.0↓) | 85.9 (9.3↓) | 87.9 (4.5↓) | 69.5 (6.9↓) | 60.2 (7.2↓) | 79.9 (6.5↓) |
| *Margin-Aware Reinforcement Learning* | | | | | | | | | |
| w/o SFT | 63.4 (2.1↓) | 43.2 (5.2↓) | 34.0 (8.3↓) | 54.5 (3.6↓) | 87.9 (7.3↓) | 88.8 (3.6↓) | 69.0 (7.5↓) | 62.7 (4.8↓) | 81.0 (5.3↓) |
| w/o MDPO | 62.8 (2.6↓) | 42.4 (6.0↓) | 31.9 (10.4↓) | 53.7 (4.4↓) | 87.1 (8.1↓) | 88.6 (3.8↓) | 70.1 (6.3↓) | 62.7 (4.8↓) | 80.9 (5.4↓) |
| MDPO→DPO | 64.6 (0.8↓) | 46.7 (1.7↓) | 37.5 (4.9↓) | 56.6 (1.4↓) | 93.5 (1.6↓) | 91.7 (0.7↓) | 74.1 (2.3↓) | 66.3 (1.2↓) | 85.1 (1.3↓) |
| MDPO→KTO | 63.1 (2.3↓) | 43.9 (4.5↓) | 34.7 (7.6↓) | 54.6 (3.5↓) | 89.1 (6.0↓) | 90.6 (1.8↓) | 68.4 (8.0↓) | 63.9 (3.6↓) | 82.2 (4.2↓) |
| MDPO→IPO | 62.5 (2.9↓) | 42.6 (5.8↓) | 32.6 (9.7↓) | 53.7 (4.4↓) | 86.3 (8.9↓) | 91.0 (1.3↓) | 65.5 (10.9↓) | 64.5 (3.0↓) | 81.3 (5.0↓) |

leads to a decline in model performance. A detailed analysis of these experiments can be found in Appendix. F.3.

# 5 LIMITATIONS, FUTURE WORK, AND CONCLUSION

**Rooted in Model-Level Research.** Frankly speaking, `LearNAT` does not achieve the best performance on BIRD. For instance, CHASE-SQL (Pourreza et al., 2024), which leverages Gemini (Butterly, 2017), attains 74.9% accuracy on BIRD-dev, outperforming `LearNAT` (32B), which achieves 65.0% (9.9%↓). Nonetheless, it is important to **clarify** that CHASE-SQL reflects a system-level solution, whereas the reported performance of `LearNAT` is evaluated strictly under a model-level setting. Although most studies do not disclose their token costs, useful comparisons can still be drawn from the limited information available. For example, CHASE-SQL reports an average token consumption of 160K[4] tokens per query, whereas `LearNAT` requires only 1.8K tokens per query. This stark contrast demonstrates that `LearNAT` is more aligned with the objectives of this study, which emphasize openness, democratization, and low-resource deployment.

**Toward System-Level Exploration.** At present, `LearNAT` has primarily focused on model-level development. In future work, we aim to further investigate the test-time scaling law of `LearNAT` to improve its performance. Appendix Fig. 10 provides a preliminary exploration of this direction, showcasing the potential of `LearNAT` at the system level. Moving forward, we plan to integrate more advanced techniques to enhance performance while minimizing token overhead, thereby systematically exploring the trade-off between accuracy and computational cost.

**Conclusion.** In this work, we propose `LearNAT`, a novel framework designed to improve the performance of LLMs on NL2SQL tasks by leveraging task decomposition and reinforcement learning. Our design is motivated by extensive preliminary experiments, through which we propose a verifiable task decomposition procedure and introduce a margin-aware DPO algorithm to optimize LLMs. The effectiveness of our approach is validated on two public NL2SQL benchmarks. Although the present study is limited to model-level improvements, our preliminary explorations demonstrate the potential of `LearNAT` at the system level. We contend that `LearNAT` represents an important step toward achieving openness, transparency, and efficiency in NL2SQL.

# 6 ETHICS STATEMENT

All datasets used for training and evaluation in this study are publicly available versions. The datasets have been curated, cleaned, and de-identified by their respective data providers prior to release. No patient personal information or identifiable medical data is present. Consequently, the research does not involve human subjects, and there are no related concerns regarding privacy, confidentiality, or legal liability.

---

[4] https://openreview.net/forum?id=CvGqMD5OtX

We strictly adhered to the usage and redistribution licenses provided by the original dataset authors and hosting platforms. Our research poses no risk of harm to individuals or groups and does not contain any potentially harmful insights, models, or applications. Additionally, there are no conflicts of interest or sponsorship concerns associated with this work. We are committed to research integrity and ethical standards consistent with the ICLR Code of Ethics.

## 7 REPRODUCIBILITY STATEMENT

We actively support the spirit of openness and reproducibility advocated by ICLR. To ensure the reproducibility of our research, we have taken the following measures:

1. Disclosure of Base Models: All base models used in our experiments are explicitly identified and described in the main manuscript. This allows readers to directly reference and obtain these models.

2. Datasets and Experimental Details: All experiments are conducted on publicly available datasets. In Appendix. D, we provide a comprehensive description of our experimental datasets. We also detail the experimental setup in Appendix. F.1. These details facilitate transparent verification and replication of our results.

3. Open-Source Code Release: To further support reproducibility, we release all training and evaluation code in a public repository (https://github.com/MrBlankness/LearNAT). The repository contains clear instructions on installation, data downloading, preprocessing, and experimentation, allowing interested researchers to replicate our results with minimal effort.

We believe that these actions align with the open science principles championed by the ICLR community, and we are committed to supporting the reproducibility and transparency of our work.

## 8 USE OF LLM

In the preparation of this manuscript, we utilized large language models (LLM) solely for writing assistance purposes. Specifically, we employed the GPT-4.1-0414 model to polish language expressions, condense sentences, and improve the overall clarity and readability of the text. The model was used exclusively for editing and refining manuscript language and did not participate in any conceptual or technical aspects of this work.

All research ideas, theoretical proof methods, experimental designs, and visualizations were conceived, executed, and finalized by the authors without the involvement of any LLM tools. The development of new concepts, formulation and validation of proofs, experimental setups, analysis of results, and the creation of figures were performed independently by the research team. At no point was the LLM model used to generate, modify, or validate the scientific content, methodology, or results presented in this article.

We emphasize that the role of GPT-4.1-0414 in this research was strictly limited to linguistic enhancement at the writing stage, and that all substantive intellectual and scientific contributions originate solely from the authors.

## ACKNOWLEDGMENTS

This work was supported by the National Natural Science Foundation of China under Grant 62576013, 62406036, and the Beijing Municipal Natural Science Foundation under Grant L251042.

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

# A    PRELIMINARY EXPERIMENTS

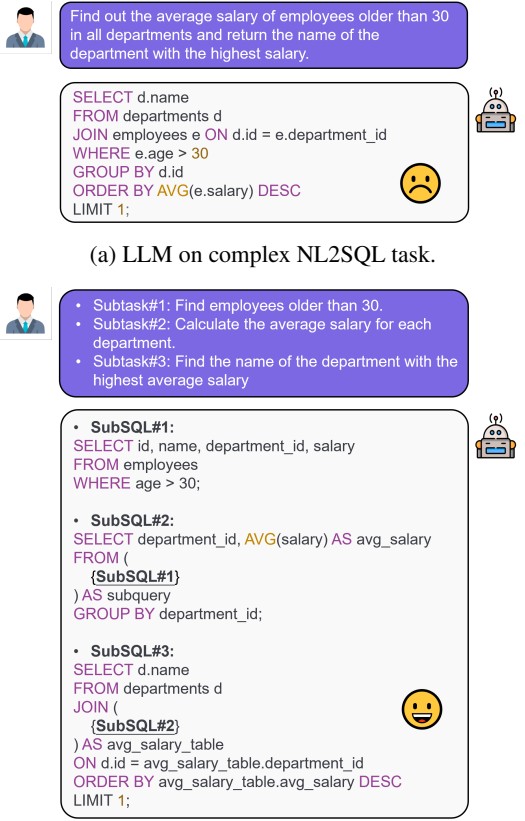

(a) LLM on complex NL2SQL task.

(b) LLM on multiple simple NL2SQL subtasks.

Figure 4: (a) illustrates the LLM directly solving a complex NL2SQL task, resulting in an incorrect output. (b) shows the LLM solving multiple decomposed simple NL2SQL subtasks from the same task in (a), resulting in a correct output. This motivates our approach to *enhancing the LLM's ability to decompose complex tasks*, thereby improving its performance on challenging NL2SQL queries.

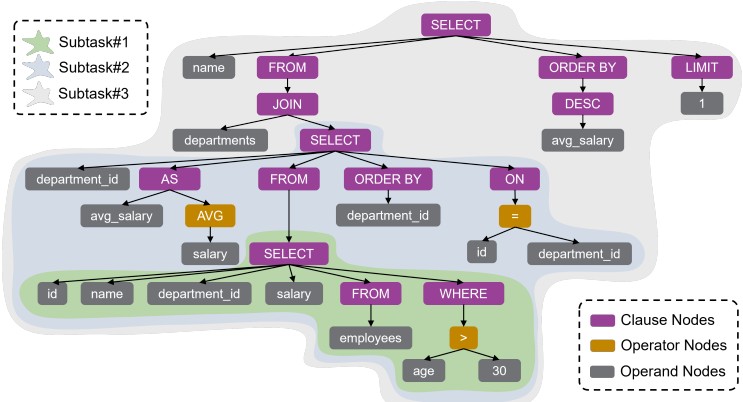

Figure 5: The abstract syntax tree (AST) of the given case in Fig. 4. Each simple NL2SQL subtask in Fig. 4 corresponds to a subtree within the AST. Clause nodes, operator nodes and operand nodes were defined in Sec. B.

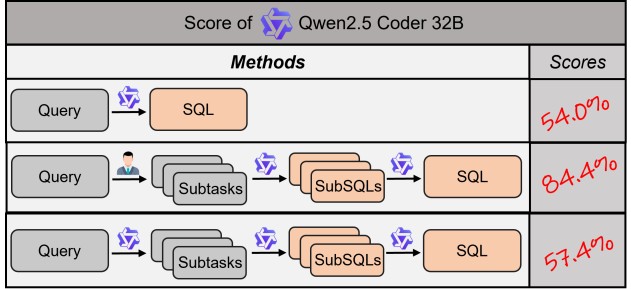

Figure 6: A preliminary experiment was conducted. We randomly selected 500 cases from the BIRD Train dataset and employed QWen-2.5-Coder to perform the NL2SQL task. The experimental results indicate that *enhancing the LLM's task decomposition ability is crucial for improving its performance on NL2SQL tasks.*

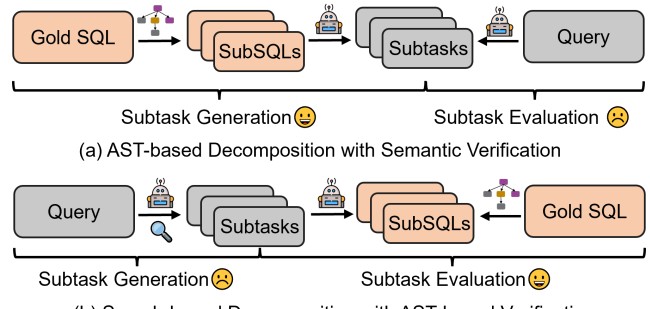

(a) AST-based Decomposition with Semantic Verification

(b) Search-based Decomposition with AST-based Verification

Figure 7: AST-based Decomposition *vs.* Search-based Decomposition

Table 5: Experimental results on evaluating subtask correctness using language models.

| Models | Accuracy | | |
|---|---|---|---|
| | Correct (227) | Error (273) | Total (500) |
| sentence-transformer | 92.1 | 9.2 | 46.8 |
| GLM-4 | 45.4 | 20.5 | 36.0 |

## B  TECHNICAL FOUNDATIONS

**Natural Language to SQL (NL2SQL).**  The goal of the NL2SQL task is to translate a natural language (NL) question $Q$ into corresponding SQL query $Y$, based on a database schema $\mathcal{DB}$. In more complex scenarios, such as those presented by BIRD (Li et al., 2023b), interpreting NL questions or understanding database values may require incorporating external knowledge, denoted by $\mathcal{K}$. The prevailing approach to the NL2SQL task adopts a cross-domain framework to assess a model's generalization ability by keeping the training, development, and test sets distinct.

**Abstract Syntax Trees (AST).**  An Abstract Syntax Tree (AST) is a structured, hierarchical representation of an SQL query, where each element of the query is captured as a node and the relationships between these elements are encoded as edges. This tree-based structure abstracts away from the linear textual representation of SQL, focusing instead on its grammatical structure and logical organization. Formally, the AST of an SQL query $Y$ can be defined as a directed acyclic graph (DAG) $\mathcal{AT}(Y) = (\mathcal{N}, \mathcal{E})$, where $\mathcal{N}$ is the set of nodes, each representing a syntactic component of the SQL query. Specifically, every node $n \in \mathcal{N}$ corresponds to a clause, operator, or operand. We categorize the nodes as follows:

Table 6: Notations of **Basic** Symbols and Their Descriptions Used in This Manuscripts.

| Symbol | Description |
|---|---|
| **Natural Language to SQL (NL2SQL)** | |
| $Q$ | Natural language (NL) question |
| $Y$ | Corresponding SQL query |
| $\mathcal{DB}$ | Database schema |
| $\mathcal{K}$ | External knowledge |
| **Abstract Syntax Trees (AST)** | |
| $\mathcal{AT}(Y) = (\mathcal{N}, \mathcal{E})$ | Abstract Syntax Tree, a directed acyclic graph of SQL query $Y$ |
| $\mathcal{N}$ | Set of nodes in AST |
| $\mathcal{E} \subseteq \mathcal{N} \times \mathcal{N}$ | Set of edges in AST |
| $n_c \in \mathcal{N}_c$ | Clause Nodes |
| $n_o \in \mathcal{N}_o$ | Operator Nodes |
| $n_v \in \mathcal{N}_v$ | Operand Nodes |
| **Monte Carlo Tree Search (MCTS)** | |
| $\mathcal{T} = (\mathcal{S}, \mathcal{A}, \mathcal{M})$ | MCTS search tree |
| $\mathcal{S}$ | Set of states or nodes in the search space |
| $\mathcal{A}(s)$ | Set of actions available at state $s$ |
| $\mathcal{M} \subseteq \mathcal{S} \times \mathcal{S}$ | Set of edges |
| $Q(s, a)$ | Estimated reward for taking action $a$ from state $s$ |
| $N(s)$ | Visit count of node $s$ |
| $c$ | Constant that controls the exploration-exploitation trade-off |
| **Direct Preference Optimization (DPO)** | |
| $x$ | Prompt |
| $y_w$ | Preferred response |
| $y_l$ | Dispreferred response |
| $p_{\mathcal{D}}^*$ | Probability of preference |
| $\pi_\theta$ | Policy model |
| $\pi_{ref}$ | Reference model |
| $\beta$ | Parameter that regulates the KL divergence |

- **Clause Nodes** ($n_c \in \mathcal{N}_c$): Represent core SQL clauses, such as `SELECT`, `FROM`, `WHERE`, `GROUP BY`, and `ORDER BY`.

- **Operator Nodes** ($n_o \in \mathcal{N}_o$): Represent logical or arithmetic operations, such as `AND`, `OR`, $=$, $>$, and $<$.

- **Operand Nodes** ($n_v \in \mathcal{N}_v$): Represent terminal elements like table names, column names, literals, or values from the database schema.

$\mathcal{E} \subseteq \mathcal{N} \times \mathcal{N}$ is the set of edges, where each directed edge $e = (n_i, n_j) \in \mathcal{E}$ captures a syntactic dependency from a parent node $n_i$ to a child node $n_j$. These edges reflect the hierarchical structure of the query, where high-level clauses dominate subcomponents or conditions.

The root node of $\mathcal{AT}(Y)$ corresponds to the main clause of the query, typically the `SELECT` clause. From the root, child nodes represent subsequent clauses or expressions, forming a hierarchical decomposition of the SQL query. For example, a `WHERE` clause node may have child nodes corresponding to individual conditions, which in turn may contain operators and operands as descendants. This formal representation enables a structured understanding of SQL queries, facilitating decomposition, syntactic validation, and step-wise reasoning. In text-to-SQL tasks, leveraging the AST structure allows efficient navigation of complex queries by guiding models through the logical and hierarchical relationships in SQL syntax.

**Monte Carlo Tree Search (MCTS).** Monte Carlo Tree Search (MCTS) is a heuristic search algorithm used for decision-making in large and complex search spaces. It combines tree-based search with stochastic sampling to balance exploration and exploitation, making it particularly effective for problems with vast or unknown state spaces. In the context of reasoning and sequential decision-

making, MCTS provides an efficient framework for discovering optimal strategies by incrementally building a search tree guided by simulation-based evaluations. Formally, MCTS operates on a search tree $\mathcal{T} = (\mathcal{S}, \mathcal{A}, \mathcal{M})$, where:

- $\mathcal{S}$ is the set of states or nodes in the search space. Each node $s \in \mathcal{S}$ represents a specific configuration of the environment, such as a partially completed plan or a subproblem in a reasoning task.
- $\mathcal{A}(s)$ denotes the set of actions available at state $s$. Each action leads to a child state $s'$, expanding the search tree.
- $\mathcal{M} \subseteq \mathcal{S} \times \mathcal{S}$ represents the set of edges, where each edge corresponds to a transition between states through an action.

The MCTS algorithm proceeds iteratively through four phases:

1. **Selection**: Starting from the root node $s_0$, the algorithm recursively selects child nodes based on a selection policy, typically using the Upper Confidence Bound for Trees (UCT) criterion:

$$a^* = \arg \max_{a \in \mathcal{A}(s)} \left( Q(s,a) + c \cdot \sqrt{\frac{\log N(s)}{N(s,a)}} \right), \tag{9}$$

   where $Q(s,a)$ is the estimated reward for taking action $a$ from state $s$, $N(s)$ is the visit count of node $s$, $N(s,a)$ is the visit count of action $a$ from $s$, and $c$ is a constant that controls the exploration-exploitation trade-off.

2. **Expansion**: If the selected node is not terminal and has unvisited child nodes, the algorithm expands the tree by adding a new child node corresponding to a valid action from the current state.

3. **Simulation (Rollout)**: From the newly expanded node, a simulation is conducted by selecting actions—often at random or based on a heuristic policy—until reaching a terminal state. The outcome of this simulation provides a reward signal, used to estimate the reward of the node.

4. **Backpropagation**: The reward obtained from the simulation is propagated back through the visited nodes, updating the reward estimations $Q(s,a)$ and visit counts $N(s,a)$ along the path from the expanded node to the root.

The output of MCTS is a policy that selects the action with the highest visit count from the root node.

$$\pi(s_0) = \arg \max_{a \in \mathcal{A}(s_0)} N(s_0, a). \tag{10}$$

**Direct Preference Optimization (DPO).** Reinforcement Learning from Human Feedback (RLHF) (Christiano et al., 2017) is an effective strategy for aligning LLMs with human preference (Ouyang et al., 2022). It relies on the Bradley-Terry (BT) model (Bradley & Terry, 1952) to define preference probability based on some reward function. Given a prompt $x$ and two responses—$y_w$ (preferred) and $y_l$ (dispreferred)—the probability of preference can be expressed as:

$$p_{\mathcal{D}}^* (y_w \succ y_l \mid x) = \sigma \left( r^*(x, y_w) - r^*(x, y_l) \right), \tag{11}$$

where $\sigma(x) = \frac{1}{1 + \exp(-x)}$ is the sigmoid function and $r^*$ represents a latent reward model. RLHF optimizes the policy model $\pi_\theta$ with a Kullback-Leibler (KL) constraint to limit deviation from a reference model $\pi_{ref}$:

$$\max \mathbb{E}_{x \sim \mathcal{D}, y \sim \pi_\theta(y|x)}[r^*(x,y)] - \beta \mathbb{D}_{KL}[\pi_\theta(y \mid x) \| \pi_{ref}(y \mid x)]. \tag{12}$$

Here, $\beta$ regulates the KL divergence to prevent reward hacking (Amodei et al., 2016). While effective, RLHF requires careful hyperparameter tuning and involves complex reward modeling and policy training.

To simplify this process, Direct Preference Optimization (DPO) (Rafailov et al., 2023) was introduced, eliminating the need for an explicit reward model. Instead, DPO directly optimizes the policy

using paired preference data. Given a prompt $x$ with responses $(y_w, y_l)$, the DPO objective maximizes the likelihood of the preferred response while minimizing that of the dispreferred one:

$$\mathcal{L}_{\text{DPO}}(\pi_\theta; \pi_{\text{ref}}) = -\mathbb{E}_{(x, y_w, y_l) \sim \mathcal{D}} \left[ \log \sigma \left( \hat{r}_\theta(x, y_w) - \hat{r}_\theta(x, y_l) \right) \right]$$
$$\hat{r}_\theta(x, y) = \beta \log \frac{\pi_\theta(y \mid x)}{\pi_{\text{ref}}(y \mid x)}. \tag{13}$$

This formulation treats $\hat{r}_\theta(x, y)$ as an "implicit reward" (Rafailov et al., 2023), allowing for direct alignment with human preference while bypassing the need for complex reward modeling and simplifying the overall training process.

## C  AST SIMILARITY

**Node-level Similarity ($\text{sim}_{\text{node}}$)**  The node-level similarity considers different types of nodes separately:

$$\text{sim}_{\text{node}}(\mathcal{AT}_1, \mathcal{AT}_2) = \sum_{t \in \{c, o, v\}} w_t \cdot \text{sim}_t(\mathcal{AT}_1, \mathcal{AT}_2), \tag{14}$$

where $w_t$ are weights for each node type with $\sum_t w_t = 1$ and $t \in \{c, o, v\}$ represents clause nodes, operator nodes, and operand nodes, respectively.

For each node type:

$$\text{sim}_t(\mathcal{AT}_1, \mathcal{AT}_2) = \frac{|\mathcal{N}_t(\mathcal{AT}_1) \cap \mathcal{N}_t(\mathcal{AT}_2)|}{|\mathcal{N}_t(\mathcal{AT}_1) \cup \mathcal{N}_t(\mathcal{AT}_2)|}, \tag{15}$$

where $\mathcal{N}_t(\mathcal{AT}_i)$ is the set of nodes of type $t$ in AST $\mathcal{AT}_i$.

**Structural Similarity ($\text{sim}_{\text{struct}}$)**  *Decomposition Synthesis Procedure* define structural similarity using the Tree Edit Distance (TED):

$$\text{sim}_{\text{struct}}(\mathcal{AT}_1, \mathcal{AT}_2) = 1 - \frac{\text{TED}(\mathcal{AT}_1, \mathcal{AT}_2)}{\max(|\mathcal{AT}_1|, |\mathcal{AT}_2|)}, \tag{16}$$

where $\text{TED}(\mathcal{AT}_1, \mathcal{AT}_2)$ is the minimum number of node operations (insertion, deletion, modification) required to transform $\mathcal{AT}_1$ into $\mathcal{AT}_2$, and $|\mathcal{AT}_i|$ is the number of nodes in AST $\mathcal{AT}_i$.

## D  DATASET STATISTICS

The BIRD and Spider datasets are introduced as follows.

- **BIRD**: BIRD (Li et al., 2023b) (Big Bench for Large-scale Database Grounded Text-to-SQL Evaluation) is a pioneering cross-domain dataset designed to assess the impact of large-scale database contents on text-to-SQL parsing. It comprises over 12,751 unique question-SQL pairs and 95 large databases with a total size of 33.4 GB, covering more than 37 professional domains, including blockchain, hockey, healthcare, and education.
- **Spider**: Spider (Yu et al., 2018) is a large-scale, cross-domain dataset for complex semantic parsing and text-to-SQL tasks, annotated by 11 Yale students. The Spider challenge aims to develop natural language interfaces for querying cross-domain databases. The dataset includes 10,181 questions paired with 5,693 unique complex SQL queries across 200 databases, spanning 138 diverse domains.

The statistics of BIRD-train, BIRD-dev, and Spider-dev used in this study are shown in Table. 7. Notably, BIRD-train does not categorize queries based on difficulty levels. Additionally, although

BIRD-train provides 9,428 data samples, the gold SQL statements for 425 of them cannot be executed by the SQL executor. Therefore, we filter out these samples considering BIRD-train to contain only 9,003 data samples in our subsequent analysis.

Table 7: Statistics for NL2SQL benchmarks.

| Benchmarks | #Queries | | | |
|---|---|---|---|---|
| BIRD-train | ~~9,428~~ 9,003 | | | |
| BIRD-dev | Simple
925 | Moderate
465 | Challenging
144 | Total
1,534 |
| Spider-dev | Easy
248 | Medium
446 | Hard
174 | Extra Hard
166 | Total
1,034 |

## E  BASELINE SOLUTIONS

We briefly describe these competitive literature used in this manuscript as follows.

System-level solutions. We compare `LearNAT` with 8 system-level methods:

- C3-SQL (Dong et al., 2023) introduces a ChatGPT-based zero-shot Text-to-SQL framework that enhances model input, mitigates model bias, and ensures output consistency through Clear Prompting, Calibration with Hints, and Consistent Output, respectively.
- DIN-SQL (Pourreza & Rafiei, 2023) introduces a task decomposition approach that improves LLMs' text-to-SQL performance by breaking query generation into sub-problems and iteratively incorporating their solutions.
- MetaSQL (Fan et al., 2024b) introduces a unified generate-then-rank framework for NLIDBs that incorporates query metadata to guide SQL generation and uses learning-to-rank algorithms to select the most accurate SQL query, improving translation accuracy across multiple benchmarks.
- MAG-SQL (Xie et al., 2024a) introduces a multi-agent generative approach with soft schema linking and iterative Sub-SQL refinement, incorporating external oversight at each generation step.
- MAC-SQL (Wang et al., 2025) introduces a multi-agent collaborative framework that combines a core decomposer agent with auxiliary agents utilizing external tools for sub-database acquisition and SQL refinement.
- SuperSQL (Li et al., 2024a) combines schema linking from RESDSQL (Li et al., 2023a), few-shot prompting and self-consistency post-processing from DAIL-SQL (Gao et al., 2024), greedy-decoding strategy from OpenAI for SQL generation, with GPT-4 as the backbone model for enhanced performance.
- SQL-o1 (Lyu et al., 2025) is a self-reward-driven, agent-based heuristic search framework for NL2SQL that employs MCTS with dynamic pruning to enable structured multi-step reasoning, significantly improving execution accuracy.
- Alpha-SQL (Li et al., 2025a) is a NL2SQL framework that leverages MCTS with an LLM-as-Action-Model to iteratively generate and refine SQL construction actions based on partial reasoning states, guided by a self-supervised reward function.

Model-level solutions. We compare `LearNAT` with 7 model-level methods:

- ACT-SQL (Zhang et al., 2023a) proposes an automatic chain-of-thought prompting method that enhances LLMs' reasoning ability in text-to-SQL tasks by leveraging schema-linking-inspired exemplars without requiring manual labeling.
- DAIL-SQL (Gao et al., 2024) proposes an integrated solution that optimizes prompt engineering methods and enhances open-source LLMs with supervised fine-tuning.
- CatSQL (Fu et al., 2023) integrates a template-based sketch with a deep learning model to improve both accuracy and runtime for NL2SQL tasks, while also proposing a Semantics Correction technique that leverages database domain knowledge to enhance the accuracy of generated queries.

- SENSE (Yang et al., 2024b) introduces a synthetic data approach that combines strong and weak model outputs for instruction tuning on open-source LLMs.
- CodeS (Li et al., 2024b) introduces a series of open-source pre-trained models, ranging from 1B to 15B parameters, specifically optimized for text-to-SQL tasks through incremental pre-training, strategic prompt construction, and bi-directional data augmentation.
- SynCoT (Liu et al., 2025) is a novel framework that enhances DPO for NL2SQL by automatically generating synthetic CoT reasoning traces to bridge the gap between preference-based learning and structured query generation, thereby unlocking DPO's full potential in this domain.
- OmniSQL (Li et al., 2025b) is trained on SynSQL-2.5M, a novel million-scale synthetic dataset featuring diverse database schemas, natural language questions, SQL queries, and chain-of-thought annotations, and it achieves state-of-the-art performance across multiple benchmarks.

MCTS-based Evaluation Protocol Explanation. Here, we provide a detailed description of the MCTS-based evaluation protocols used in our comparative experiments.

- **Evaluation protocols of SQL-o1**: SQL-o1 employs MCTS during inference. Specifically, at each node of the MCTS tree, SQL-o1 generates a subtask along with its corresponding SQL statement, a mechanism closely resembling that of `LearNAT`. The complete task chain relevant to the user query is defined by the trajectory from the root node to a leaf node, based on which the final SQL query is constructed. Furthermore, SQL-o1 defines a reward for each node by evaluating the model's confidence—i.e., the probability assigned by the LLM to the output at that node—and selects the node with the highest confidence. Nodes whose confidence falls below a predefined threshold are pruned and not further expanded.
- **Evaluation protocols of Alpha-SQL**: Alpha-SQL also integrates MCTS during inference. However, unlike SQL-o1, Alpha-SQL does not decompose the problem into explicit subtasks; instead, each node directly corresponds to the user query and an associated SQL statement. Consequently, while SQL-o1 defines its action space as the generation of the next planning step and its corresponding SQL, Alpha-SQL adopts a richer set of atomic actions: *Rephrase Question*, *Schema Selection*, *Column Value Identification*, *Column Function Identification*, *SQL Generation*, *SQL Revision*, and *Termination*. Through sequential selection of these actions, Alpha-SQL iteratively refines both the natural language query representation and the corresponding SQL. Alpha-SQL also defines node rewards based on the consistency frequency of paths obtained via high-temperature sampling over action sequences. Upon completion of the MCTS search, Alpha-SQL aggregates all SQL queries generated across trajectories and selects the final output based on self-consistency, i.e., agreement among execution results of candidate SQL queries.

# F  ADDITIONAL EXPERIMENTS

## F.1  IMPLEMENTATION DETAILS.

We employ GLM-4-Plus[5] as the primary model for synthesizing decomposition data and fine-tune the model on Qwen2.5-Coder (Yang et al., 2024a), including its 7B, 14B, and 32B versions. We used the PyTorch library to implement all the algorithms based on the open-source HuggingFace transformers (Wolf et al., 2019) and LLaMA-Factory (Zheng et al., 2024). The experiments are conducted on 8×A100 GPUs. During the SFT stage, we utilize the AdamW optimizer with a learning rate of 2e-5 and a cosine warmup scheduler over three epochs. For DPO training, the Adam optimizer is used with a learning rate of 2e-6, and the $\beta$ parameter is set to 0.2, in accordance with the original DPO configuration. In Eq. 14, we assign equal weights to all three nodes, i.e., $w_c = w_o = w_v = 0.33$. Based on our experimental observations F.12, we set $\alpha = 0.75$ in Eq. 5. During inference, we strictly follow the evaluation protocol provided by DAIL-SQL (Gao et al., 2024) (the Without Voting setting). The protocol provides a complete set of prompts to better structure the instructions, user queries, and database schema information, enabling the LLM to generate a single response from which the SQL statement is extracted as the final answer.

We adopt the following strategy to adapt `LearNAT` to MCTS-based evaluation protocols.

---

[5]https://bigmodel.cn/dev/api/normal-model/glm-4

- **Adapt `LearNAT` to SQL-o1**: When adapting LearNAT to the SQL-o1 framework, we leverage LearNAT to generate both subtasks and subSQLs at each MCTS node. The tree traversal and pruning decisions are guided by the same reward mechanism used in SQL-o1, and the path with the highest cumulative confidence is ultimately selected as the final SQL query.
- **Adapt `LearNAT` to Alpha-SQL**: In migrating LearNAT to the Alpha-SQL setting, we adopt a straightforward strategy: every LLM invocation within Alpha-SQL is replaced with LearNAT's model parameters. Notably, for the *SQL Generation* action, we retain LearNAT's original prompting scheme to ensure that SQL is still produced in a subtask-by-subtask manner. For all other actions, we follow Alpha-SQL's original prompting design.

### F.2 ERROR CASE ANALYSIS

To further investigate the reasons for the failure of the *Decomposition Synthesis Procedure* in certain cases, we randomly selected 50 unsuccessful cases for error analysis. The error distribution is shown in Fig. 3. The results indicate that the decomposition failures can be attributed to four distinct types of errors, including schema linking, float computation, unknown rules, and error answer.

We analyze these errors one by one by presenting typical cases for each of the four error attributions.

**Case for Schema Linking.**

[#Question:] *What is the user avatar url for user 41579158? What is the latest movie rated by him / her?*

[#Evidence:] *user avatar url refers to* `user_avatar_image_url`; *latest movie rated refers to latest* `rating_date`;

[#Gold SQL]

```sql
SELECT T3.user_avatar_image_url, T3.rating_date_utc
FROM movies AS T1
INNER JOIN ratings AS T2 ON T1.movie_id = T2.movie_id
INNER JOIN ratings_users AS T3 ON T3.user_id = T2.user_id
WHERE T3.user_id = 41579158
ORDER BY T3.rating_date_utc DESC
LIMIT 1
```

[#Predict SQL]

```sql
SELECT user_avatar_image_url, movie_id FROM (
SELECT T3.user_avatar_image_url, T3.rating_date_utc FROM (
SELECT T2.user_id, T2.movie_id FROM ratings AS T2
WHERE T2.user_id = 41579158
) AS Sub1
INNER JOIN ratings_users AS T3
ON Sub1.user_id = T3.user_id
WHERE T3.user_id = 41579158
) AS Sub2
ORDER BY rating_date_utc DESC
LIMIT 1;
```

In this case, the LLM misidentified the column, mapping "*the latest movie rated by him/her*" to the `movie_id` column instead of the `rating_date_utc` column. However, the evidence provided relevant information (although it did not explicitly specify the corresponding column).

**Case for Float Computation.**

[#Question:] *What is the percentage of the ratings were rated by user who was a subscriber?*

[#Evidence:] *user is a subscriber refers to* `user_subscriber = 1`; *percentage of* `ratings = DIVIDE(SUM(user_subscriber = 1), SUM(rating_score))` *as percent;*

[#Gold SQL]

```sql
SELECT (CAST(SUM(
CASE WHEN user_subscriber = 1 THEN 1 ELSE 0 END
```

```
) AS REAL) * 100 / COUNT(*)
) FROM ratings
```

[#Predict SQL]

```
SELECT (CAST(SUM(
CASE WHEN user_subscriber = 1 THEN 1 ELSE 0 END
) AS REAL) / COUNT(*) * 100
) FROM ratings
```

In this case, the LLM did not strictly follow the Gold SQL in executing multiplication before division but instead generated SQL that performed the operations in the reverse order. Although mathematically equivalent, floating-point arithmetic in SQL can introduce numerical precision variations. Since our evaluation metric is Execution Accuracy, this discrepancy led to an inconsistency in the results. Specifically, the Gold SQL produced an execution result of 21.648420738414252, whereas the Predicted SQL yielded 21.64842073841425.

**Case for Unknown Rules.**

[#Question:] *List all movies with the best rating score. State the movie title and number of Mubi user who loves the movie.*

[#Evidence:] *best rating score refers to* `rating_score = 5`; *number of Mubi user who loves the movie refers to* `movie_popularity`

[#Gold SQL]

```
SELECT DISTINCT T2.movie_title, T2.movie_popularity
FROM ratings AS T1 INNER JOIN movies AS T2
ON T1.movie_id = T2.movie_id
WHERE T1.rating_score = 5
```

[#Predict SQL]

```
SELECT T2.movie_title, T2.movie_popularity
FROM ratings AS T1 INNER JOIN movies AS T2
ON T1.movie_id = T2.movie_id
WHERE T1.rating_score = 5
```

In this case, the Gold SQL performed an additional deduplication step (`DISTINCT`) on the query results, whereas the Predicted SQL did not. This deduplication is a default user-friendly operation, but it was not explicitly stated in the query. As a result, the execution results of the Predicted SQL and Gold SQL differed.

**Case for Error Answer.**

[#Question:] *What is the name of the longest movie title? When was it released?*

[#Evidence:] *longest movie title refers to* `MAX(LENGTH(movie_title))`; *when it was released refers to* `movie_release_year`

[#Gold SQL]

```
SELECT movie_title, movie_release_year FROM movies
ORDER BY LENGTH(movie_popularity) DESC
LIMIT 1
```

[#Predict SQL]

```
SELECT movie_title, movie_release_year FROM movies
WHERE LENGTH(movie_title) = (
    SELECT MAX(LENGTH(movie_title)) FROM movies
)
```

Some cases in BIRD-train contain incorrect Gold SQL. For example, in this case, the query requires computing the longest movie, and the evidence explicitly states that the correct computation should

be `MAX(LENGTH(movie_title))`. However, the Gold SQL incorrectly calculates this by using `LENGTH(movie_popularity)`, which is clearly incorrect. In contrast, the Predicted SQL correctly implements the intended computation. Therefore, the decomposition failure in this case is a false negative, caused by an error in the Gold SQL.

## F.3 ABLATION STUDIES

**Effectiveness of `LearNAT`.** First, we present the most naive baseline (w/o `LearNAT`), which represents the basic performance of Qwen2.5-Coder-7B. This experiment demonstrates the strong performance of `LearNAT`, which significantly boosts a vanilla Qwen2.5-Coder-7B model. For instance, it yields a remarkable 9.4%↑ on the Spider-dev set and a 10.6%↑ on the BIRD dataset. To validate the effectiveness of task decomposition, we replace `LearNAT` with a naive DPO algorithm—i.e., applying a simple reinforcement learning strategy without incorporating any decomposition mechanisms. Experimental results show that this baseline performs significantly worse than `LearNAT`, achieving only 79.0% (7.4↓) accuracy on Spider-dev and 52.8% (5.3↓) on BIRD-dev. This substantial performance gap highlights the critical role of task decomposition in solving complex NL2SQL tasks. In addition, we conduct a simple experiment using naive Qwen2.5-Coder-7B with CoT-based decomposition, where the LLM directly decomposes the NL2SQL task and generates SQL. While this setup improves performance (e.g., 1.8%↑ on BIRD-dev), it is far less effective than `LearNAT`, highlighting the importance of AST-guide decomposition, reinforcement learning and adaptive demonstrations.

**Effectiveness of Decomposition Synthesis Procedure.** We remove the AST-guide, replacing it with naive MCTS for decomposition and using vanilla DPO in reinforcement learning. The results show an improvement over w/o `LearNAT` (e.g., 5.6%↑ on BIRD-dev), indicating that decomposition-based RL enhances LLM performance in complex NL2SQL tasks. However, compared to `LearNAT`, the model's performance drops significantly (e.g., 5.0%↓ on BIRD-dev), suggesting that without an appropriate reward evaluation, performance improvements are limited. `LearNAT` tightly integrates reward modeling with AST, designing a rule-based reward model that significantly enhances LLM performance.

**Effectiveness of Margin-Aware Reinforcement Learning.** We remove the SFT stage, leading to a performance drop (e.g., 3.6%↓ on BIRD-dev), indicating that SFT is necessary for initializing the LLM before applying MDPO, aligning with findings from prior work (Yang et al., 2024b). Similarly, removing MDPO results in a performance decline (e.g., 4.4%↓ on BIRD-dev), showing that SFT alone teaches the LLM to generate correct outputs but fails to suppress incorrect ones (Liao et al., 2024), which degrades overall model performance. Replacing MDPO with naive DPO further reduces performance, as the lack of margin awareness prevents the LLM from distinguishing critical steps during preference learning, leading to coarse-grained reward estimation and thus suboptimal performance.

## F.4 INFERENCE TIME COMPARISON BETWEEN LEARNAT AND SYSTEM-LEVEL METHODS

We summarize the inference-time overhead of `LearNAT` and System-Level Methods in the Table. 8. The experimental results demonstrate that, while achieving comparable performance, `LearNAT` achieves a substantially lower inference-time cost.

Table 8: Inference Time Comparison Between `LearNAT` and System-Level Methods. The inference time of `LearNAT` is normalized to 1.

| Methods | C3-SQL | DIN-SQL | MetaSQL | Mag-SQL | SuperSQL | MAC-SQL | SQL-o1 | Alpha-SQL | LearNAT |
|---------|--------|---------|---------|---------|----------|---------|--------|-----------|---------|
| *Inference Time* | 11.8 | 5.4 | 5.6 | 6.3 | 5.1 | 3.6 | 5.3 | 111.2 | 1.0 |

## F.5 ANALYSIS OF SQL DIVERSITY

We note that, during the MCTS-based subtask search process, some potentially correct subtasks are indeed generated. However, because the ASTs of these subtasks do not exist as subtrees in the AST of the ground-truth SQL, they are mistakenly judged as incorrect trajectories. To address this, we designed the following experiment: specifically, we use GLM-4-Plus to rewrite the SQL for

each query in the training data. The rewriting is constrained such that the AST structures of the original and rewritten SQL are different, but their execution results are identical, thereby enhancing the diversity of the synthesized data. We denote this method as `LearNAT`$^+$. Our experimental results, as shown in Table. 9, indicate that increasing data diversity in this way can further improve the performance of `LearNAT`.

Table 9: Performance comparison of `LearNAT` and `LearNAT`$^+$ with enhanced SQL diversity. **Bold** indicates the better result.

| Methods | BIRD-dev (In-Domain) | | | | Spider-dev (Out-of-Domain) | | | | |
|---|---|---|---|---|---|---|---|---|---|
| | Simple | Moderate | Challenging | Total | Easy | Medium | Hard | Extra Hard | Total |
| Qwen2.5-Coder-7B | | | | | | | | | |
| LearNAT | 65.4 | 48.4 | 42.4 | 58.1 | 95.2 | 92.4 | 76.4 | 67.5 | 86.4 |
| LearNAT$^+$ | **66.6** | **49.5** | **46.5** | **59.5** | **96.8** | **93.3** | **83.3** | **68.7** | **88.5** |

We think that the strict AST-based evaluation in `LearNAT` does indeed reduce the diversity of the model's outputs. However, while enhancing the diversity of model outputs is certainly a desirable goal, we believe it is not the most critical one. Our primary motivation behind `LearNAT` is to teach the model to reason correctly subtask by subtask in this work. The strict AST-based supervision guarantees that the synthesized decompositions form a valid sequence of subtasks. This guarantee of correctness is the key driver of `LearNAT`'s performance gains.

## F.6 ANALYSIS OF LEARNAT IN SEMI-SUPERVISED SETTINGS

We believe `LearNAT` is also suitable for semi-supervised scenarios. As a concrete example, we assume the following semi-supervised scenario: we have a labeled dataset BIRD-train-part1 and an unlabeled dataset BIRD-train-part2, each comprising half of the full BIRD-train dataset. We perform the following semi-supervised learning procedure:

1. We use `LearNAT` to construct task-decomposition data on BIRD-train-part1, where the correctness of each subtask is verified using the ground-truth AST.
2. Using the synthesized data from Step 1, we fine-tune the Qwen2.5-Coder-7B model (denoted as `LearNAT`$^1$).
3. We then apply the fine-tuned model from Step 2 to construct task-decomposition data on BIRD-train-part2, without verifying subtask correctness using a ground-truth AST.
4. We combine the synthetic data from Steps 1 and 3, and use this merged dataset to fine-tune the original Qwen2.5-Coder-7B model, yielding `LearNAT`$^2$.

The performance of these models is shown in Table. 10:

Table 10: Performance of `LearNAT` in semi-supervised settings.

| Methods | BIRD | | | | Spider | | | | |
|---|---|---|---|---|---|---|---|---|---|
| | Simple | Moderate | Challenging | Total | Easy | Medium | Hard | Extra Hard | Total |
| Qwen2.5-Coder-7B | 56.1 | 34.5 | 33.8 | 47.5 | 82.7 | 84.1 | 71.8 | 54.8 | 77.0 |
| LearNAT$^1$ | 61.4 | 42.2 | 38.9 | 53.5 | 90.3 | 88.8 | 73.6 | 62.0 | 82.3 |
| LearNAT$^2$ | 62.1 | 45.2 | 38.9 | 54.8 | 92.3 | 89.9 | 74.1 | 62.0 | 83.4 |

The above experiment demonstrates the effectiveness of `LearNAT` in a semi-supervised setting. `LearNAT` can rely entirely on a small labeled dataset with ground-truth SQL to initialize the model, and then use the capabilities acquired from this small dataset to annotate the unlabeled portion, thereby reducing dependence on labeled data. However, it is important to note that `LearNAT`$^2$ achieves only 54.8% on the BIRD dataset, whereas `LearNAT` trained with the fully labeled dataset reaches 58.1%. Thus, although `LearNAT` proves its feasibility in semi-supervised scenarios, the supervised setting remains the more suitable and effective regime for `LearNAT`.

### F.7 COMPARISON OF TOKEN COST

We follow SuperSQL (Li et al., 2024a) and report the *Avg. Tokens (k) / Query* and *EX / Avg. Tokens (k)* metrics for `LearNAT` and baseline methods. The experimental results are presented in Table. 11.

Table 11: Token Cost Comparison of `LearNAT` and baseline methods. **Bold** indicates the best result, while underline denotes the second-best results.

| Methods | LLMs | BIRD | Spider | Total | Avg. Tokens (k) / Query | EX / Avg. Tokens (k) |
|---------|------|------|--------|-------|--------------------------|----------------------|
| | | | | System-Level | | |
| C3-SQL | GPT-4 | 50.2 | 82.0 | 63.0 | 21.2 | 3.0 |
| DIN-SQL | GPT-4 | 50.7 | 74.2 | 60.2 | 9.7 | 6.2 |
| MetaSQL | GPT-4 | 47.6 | 69.6 | 56.5 | 10.1 | 5.6 |
| MAG-SQL | GPT-4 | 57.6 | 85.3 | 68.8 | 11.4 | 6.0 |
| SuperSQL | GPT-4 | 58.5 | 87.0 | 70.0 | 9.1 | 7.7 |
| MAC-SQL | GPT-4 | 59.4 | 86.7 | 70.4 | 6.5 | 10.8 |
| | | | | Model-Level | | |
| ACT-SQL | GPT-4 | 52.4 | 74.5 | 61.3 | 1.6 | 38.3 |
| DAIL-SQL | GPT-4 | 54.3 | 76.2 | 63.1 | 1.3 | 48.6 |
| CodeS | CodeS-7B | 57.0 | 85.4 | 68.4 | **1.1** | 62.2 |
| | CodeS-15B | 58.5 | 84.9 | 69.1 | **1.1** | **62.8** |
| | | | | Ours | | |
| LearNAT | Qwen2.5-Coder-7B | 58.1 | 86.4 | 69.5 | 1.8 | 38.6 |
| | Qwen2.5-Coder-14B | 61.2 | 86.9 | 71.6 | 1.7 | 42.1 |
| | Qwen2.5-Coder-32B | **65.0** | **88.4** | **74.4** | 1.8 | 41.3 |

The experimental results show that, compared with system-level methods, `LearNAT` achieves substantial advantages in terms of performance, token consumption, and performance per token. When compared with model-level methods, although `LearNAT` attains higher performance, it consumes more tokens. This is because `LearNAT` output both the intermediate subtasks and their corresponding SQL statements, which leads to increased token usage.

### F.8 ROBUSTNESS ANALYSIS OF LEARNAT

In Table. 12, we provide the mean and standard deviation of three versions of the `LearNAT` model over five experimental runs, to ensure the reliability and robustness of `LearNAT`'s performance.

Table 12: Robustness analysis of `LearNAT`, the mean and standard deviation are reported.

| LLMs | BIRD-dev (In-Domain) | | | | Spider-dev (Out-of-Domain) | | | | |
|------|--------|----------|-------------|-------|------|--------|------|------------|-------|
| | Simple | Moderate | Challenging | Total | Easy | Medium | Hard | Extra Hard | Total |
| Qwen2.5-Coder-7B | 65.1±1.2 | 47.9±1.4 | 41.4±3.1 | 57.6±0.7 | 94.5±2.1 | 92.6±0.9 | 76.6±2.2 | 66.6±1.0 | 86.2±0.4 |
| Qwen2.5-Coder-14B | 68.3±0.8 | 51.4±0.9 | 44.6±1.0 | 61.0±0.3 | 95.8±2.4 | 91.6±1.2 | 80.6±3.7 | 68.0±1.4 | 86.9±0.3 |
| Qwen2.5-Coder-32B | 70.6±0.7 | 55.4±1.0 | 58.3±1.4 | 64.8±0.3 | 96.0±2.1 | 92.5±1.0 | 84.6±2.2 | 69.3±1.5 | 88.3±0.2 |

### F.9 CASE ANALYSIS BETWEEN DPO AND MARGIN-AWARE DPO

In this subsection, we further discuss the insights behind our margin-aware DPO design and believe that clarifying the idea will strengthen the manuscript. Concretely, consider the following case:

[#Question:] *Consider the average difference between K-12 enrollment and 15-17 enrollment of schools that are locally funded, list the names and DOC type of schools which has a difference above this average.*

[#Gold SQL]

```
SELECT T2.School, T2.DOC
FROM frpm AS T1 INNER JOIN schools AS T2
ON T1.CDSCode = T2.CDSCode
WHERE T2.FundingType = 'Locally_funded'
  AND (T1.'Enrollment (K-12)' - T1.'Enrollment (Ages 5-17)') > (
```

```
    SELECT AVG(T3.'Enrollment (K-12)' - T3.'Enrollment (Ages 5-17)')
    FROM frpm AS T3 INNER JOIN schools AS T4
    ON T3.CDSCode = T4.CDSCode
    WHERE T4.FundingType = 'Locally_funded'
)
```

In the NL2SQL task decomposition for this example, we obtain the following correct subtask and sub-SQL (denoted Response #1):

[#Subtask:] *Compute the average difference between K–12 enrollment and 15–17 enrollment for locally funded schools.*

[#SubSQL]

```
SELECT AVG(T1.'Enrollment (K-12)' - T1.'Enrollment (Ages 5-17)') AS avg_diff
FROM frpm AS T1
INNER JOIN schools AS T2
  ON T1.CDSCode = T2.CDSCode
WHERE T2.FundingType = 'Locally_funded';
```

We also collected two erroneous responses. The first erroneous response (Response #2) is:

[#Subtask:] *Compute the average difference between K–12 enrollment and 15–17 enrollment.*

[#SubSQL]

```
SELECT AVG(T1.'Enrollment (K-12)' - T1.'Enrollment (Ages 5-17)') AS avg_diff
FROM frpm AS T1
INNER JOIN schools AS T2
  ON T1.CDSCode = T2.CDSCode;
```

The second erroneous response (Response #3) is:

[#Subtask:] *Compute the average difference between K–12 enrollment and 15–17 enrollment for locally funded schools.*

[#SubSQL]

```
SELECT AVG(T1.'Enrollment (K-12)' - T1.'Enrollment (Ages 5-17)') AS avg_diff
FROM frpm AS T1
INNER JOIN schools AS T2
  ON T1.CDSCode = T2.CDSCode
WHERE T2.FundingType = 'Locally';
```

We observe that Response #2 is a more severe error than Response #3: Response #3 only uses an incorrect *FundingType* value (a string-level mistake), whereas Response #2 omits the funding filter entirely and thus computes the average across all funding types, yielding a fundamentally incorrect statistic.

However, in standard DPO optimization, Response #2 and Response #3 are treated as equivalent rejected samples, despite exhibiting clearly different degrees of error. We argue that such differences should be explicitly distinguished. Therefore, we introduce Margin-aware DPO, which incorporates an AST-based metric as an offset term in the DPO loss to differentiate rejected samples according to their error severity.

Under this offset mechanism, the reward margin between Response #1 and Response #2 becomes larger than that between Response #1 and Response #3. We believe that dynamically adjusting the reward margin between the chosen sample and rejected samples of varying error levels enables the model to better distinguish among incorrect responses, thereby allowing it to perform more fine-grained preference optimization.

### F.10 TRAINING ANALYSIS OF MDPO

In Fig. 8, we present the training loss curves of both DPO and MDPO. Based on the comparison of training losses, we observe that MDPO exhibits a faster and more pronounced training loss reduction compared with DPO. This observation indicates that the reward margin introduced in MDPO

Table 13: Performance of LearNAT on various Backbone LLMs. The red font indicates the performance improvement caused by using LearNAT.

| Methods | BIRD-dev (In-Domain) | | | | Spider-dev (Out-of-Domain) | | | | |
|---|---|---|---|---|---|---|---|---|---|
| | Simple | Moderate | Challenging | Total | Easy | Medium | Hard | Extra Hard | Total |
| Qwen2.5-Coder-7B | | | | | | | | | |
| | 56.1 | 34.5 | 33.8 | 47.5 | 82.7 | 84.1 | 71.8 | 54.8 | 77.0 |
| LearNAT | 65.4 (9.3↑) | 48.4 (13.9↑) | 42.4 (8.6↑) | 58.1 (10.6↑) | 95.2 (12.5↑) | 92.4 (8.3↑) | 76.4 (4.6↑) | 67.5 (12.7↑) | 86.4 (9.4↑) |
| Qwen2.5-Coder-14B | | | | | | | | | |
| | 59.5 | 42.6 | 38.4 | 52.4 | 86.6 | 86.4 | 72.9 | 55.6 | 79.2 |
| LearNAT | 68.5 (9.0↑) | 51.4 (8.8↑) | 45.8 (7.4↑) | 61.2 (8.8↑) | 95.6 (9.0↑) | 91.5 (5.1↑) | 80.5 (7.6↑) | 68.7 (13.1↑) | 86.9 (7.7↑) |
| Qwen2.5-Coder-32B | | | | | | | | | |
| | 64.8 | 47.8 | 41.4 | 57.4 | 93.5 | 87.7 | 75.9 | 58.4 | 82.4 |
| LearNAT | 70.7 (5.9↑) | 55.5 (7.7↑) | 59.0 (17.6↑) | 65.0 (7.6↑) | 96.4 (2.9↑) | 92.4 (4.7↑) | 85.1 (9.2↑) | 69.3 (10.9↑) | 88.4 (6.0↑) |
| GLM4-9B | | | | | | | | | |
| | 55.5 | 36.3 | 25.0 | 46.8 | 82.3 | 85.9 | 62.6 | 59.6 | 76.9 |
| LearNAT | 64.0 (8.5↑) | 41.9 (5.6↑) | 31.3 (6.3↑) | 54.2 (7.4↑) | 89.1 (6.8↑) | 90.1 (4.2↑) | 71.8 (9.2↑) | 65.1 (5.5↑) | 82.8 (5.9↑) |
| Meta-Llama-3-8B-Instruct | | | | | | | | | |
| | 56.5 | 37.0 | 25.0 | 47.7 | 83.9 | 85.4 | 64.4 | 56.6 | 76.9 |
| LearNAT | 64.9 (8.4↑) | 42.6 (5.6↑) | 40.3 (15.3↑) | 55.8 (8.1↑) | 90.7 (6.8↑) | 91.7 (6.3↑) | 71.3 (6.9↑) | 63.9 (7.3↑) | 83.6 (6.7↑) |

provides larger and more meaningful gradients for model optimization, which can be attributed to MDPO's ability to exploit the reward differences among negative samples.

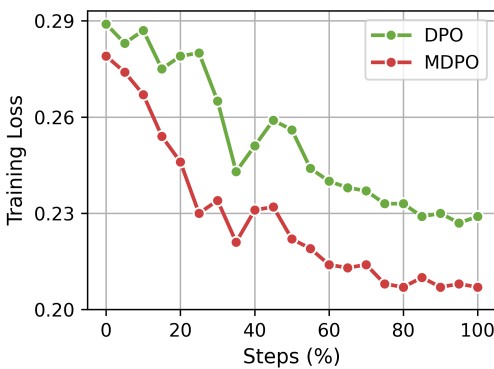

Figure 8: Comparison of the trend of training loss between DPO and MDPO.

## F.11 ANALYSIS ON VARIOUS BACKBONE LLMS

Table. 13 presents the performance of LearNAT across backbone LLMs of varying sizes. The results reveal several key observations: (1) LearNAT consistently improves performance across different model sizes. (2) As the number of parameters in the backbone LLM increases—for instance, from 7B to 14B to 32B—the inherent NL2SQL capability of the model improves accordingly, and this trend is also reflected in the performance gains achieved by LearNAT. (3) Notably, LearNAT enables smaller models to outperform significantly larger ones, effectively mitigating the limitations imposed by model scale. For example, after training with LearNAT, Qwen2.5-Coder-7B achieves 86.4% on Spider-dev, surpassing the naive Qwen2.5-Coder-32B, which achieves only 82.4% (4.0%↓). A similar trend is observed on the BIRD-dev dataset.

To further demonstrate the generality and robustness of LearNAT, we incorporate LLMs with different architectures, such as GLM4-9B and Meta-Llama-3-8B-Instruct, as backbones. The results show that LearNAT consistently yields substantial gains across architectures—for instance, improving GLM4-9B by 7.4%↑ and Meta-Llama-3-8B-Instruct by 8.1%↑ on BIRD-dev. These results confirm that LearNAT is both architecture-agnostic and highly effective across a wide range of LLM configurations.

## F.12 ANALYSIS OF AST SIMILARITY

We evaluate the importance of node similarity and structural similarity in `LearNAT` by adjusting the weight parameter $\alpha$ in Eq. 5. Specifically, we vary $\alpha$ between 0, 0.25, 0.5, 0.75.

Experimental results (illustrated in Fig. 9) show that using only node similarity or only structural similarity leads to performance degradation, indicating that both types of similarity contribute to evaluation quality. A balanced setting ($\alpha = 0.5$) does not achieve optimal performance. `LearNAT` achieves the best performance when $0.5 < \alpha < 1$, suggesting that node similarity is more effective than structural similarity in AST-based similarity assessment. This highlights that while both node and structural similarity are necessary, node similarity plays a slightly more critical role in guiding AST-based decomposition and reward estimation.

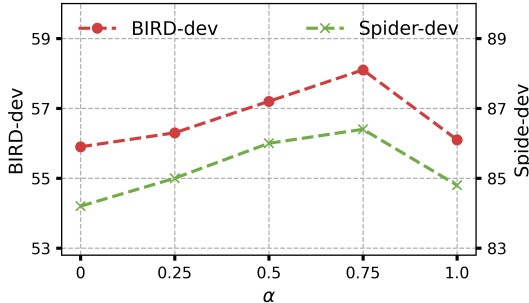

Figure 9: Execution accuracy on BIRD-dev and Spider-dev using various $\alpha$ in AST similarity estimation.

## F.13 SYSTEM LEVEL POTENTIAL OF LEARNAT

We have conducted new experiments to explore the performance of `LearNAT` under a system-level setting. The results are presented in Fig. 10. In this setup, we invoke `LearNAT` multiple times for the same query to generate a set of candidate SQL, and then apply the most basic form of self-consistency to select the final SQL output. Results demonstrate that `LearNAT` can indeed benefit from multi-call to improve performance; however, this improvement comes at the cost of significantly increased token consumption.

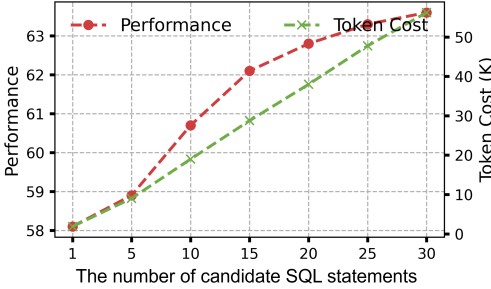

Figure 10: Performance and Token Cost of `LearNAT` with Qwen2.5-Coder-7B as backbone LLM using generating multiple candidate SQL and selecting SQL via self-consistency.

## G RELATED WORK

### G.1 NL2SQL PARSING BASED ON LLMS

**Model-level Solution.** Model-level solutions refer to approaches where a single large language model, given a natural language query, database schema, and optionally additional instructions, generates a single SQL statement in an end-to-end manner. This SQL is directly used as the final output.

Model level solutions are typically based on model fine-tuning methods. Model fine-tuning (Zhang et al., 2023b) adapts pre-trained LLMs to specific tasks by adjusting model parameters through additional training. While promising for NL2SQL, this approach is limited to public models with accessible parameters. Due to the performance gap between large-scale private and small-scale public models, existing research has primarily focused on system-level solution, with relatively few studies (Yang et al., 2024b; Wu et al., 2024; Li et al., 2024b; Sun et al., 2023; Li et al., 2023a) dedicated to fine-tuning open-source models. *Despite their empirical success, these studies focus solely on learning the target SQL queries while neglecting the reasoning process involved in parsing complex SQL structures. This results in mere memorization of outcomes rather than fostering a deep understanding of the underlying problems.*

The works most closely related to ours are Reasoning-SQL (Pourreza et al., 2025), SynCoT (Liu et al., 2025), and Struct-LLM (Stoisser et al., 2025). These approaches similarly incorporate reasoning during the inference stage and employ reinforcement learning algorithms to enhance the model's reasoning capabilities. Both Reasoning-SQL (Pourreza et al., 2025) and Struct-LLM (Stoisser et al., 2025) optimize LLM performance on NL2SQL by replacing naïve binary rewards with continuous reward signals. However, although they design various reward mechanisms, all of these rewards are applied only to the final generated SQL. In contrast, the rewards designed in `LearNAT` are primarily used to evaluate intermediate subtasks. In other words, *`LearNAT` introduces a **process reward** (Wang et al., 2024; Lightman et al., 2024) model that explicitly regulates the correctness of intermediate reasoning steps, whereas Reasoning-SQL and Struct-LLM focuses on **outcome rewards** (Wang et al., 2024; Lightman et al., 2024).*

**System-level Solution.** System-level methods go beyond the end-to-end generation by the LLM. These approaches typically incorporate additional components such as schema linking, candidate SQL generation, result selection or consistency verification, and SQL refinement, aiming to improve overall robustness and accuracy. System-level solutions are typically based on Prompt engineering. Prompt engineering (Ekin, 2023) aims to guide model outputs towards desired results through carefully designed input prompts and can be applied to both open-source and proprietary models. In the NL2SQL domain, prompt engineering serves as a crucial technique for enhancing the performance of LLMs (Kim et al., 2020; Li et al., 2024a). Several studies (Gao et al., 2024; Mao et al., 2024; Dong et al., 2023; Lee et al., 2025; Talaei et al., 2024; Pourreza et al., 2024) have explored different prompt engineering strategies to enhance NL2SQL performance. The most relevant works are DIN-SQL (Pourreza & Rafiei, 2023) and MAC-SQL (Wang et al., 2025), which employ zero-shot prompting (*Let's think step by step*) or few-shot prompting (e.g., using a small set of demonstrations) to help LLMs decompose complex NL2SQL tasks. While these methods have achieved significant success on publicly available NL2SQL benchmarks, *open-source models, constrained by smaller parameter sizes and limited pretraining knowledge, exhibit substantially weaker performance in task decomposition compared to closed-source models (Shen et al., 2023).*

## G.2 Enhancing Reasoning with RL

**Search-Guided Reasoning in LLMs.** Recent research efforts (Feng et al., 2023; Chen et al., 2024; Xie et al., 2024b) aiming at advancing the reasoning capabilities of LLMs have increasingly incorporated Monte Carlo Tree Search to generate trajectories for model training, yielding significant improvements in reasoning performance. Despite these successes, MCTS-driven methods still face several challenges, such as the *vast search space* inherent to language models and the *difficulty of quantifying node rewards*. Existing research in the mathematical domain primarily relies on self-evaluation or training external evaluation models based on labeled data. In the NL2SQL domain, *we introduce a novel approach that leverages abstract syntax trees to quantify node rewards, effectively guiding the model to prioritize the exploration of the most valuable nodes.*

**Direct Preference Optimization (DPO) Algorithms.** Among various reinforcement learning algorithms, Direct Preference Optimization (DPO) (Rafailov et al., 2023) has gained popularity due to its simplicity. DPO relies on instance-level preference signals for model optimization. However, it faces challenges in handling multi-step reasoning tasks, as it struggles to rectify specific errors that arise during the reasoning process (Hwang et al., 2024; Liao et al., 2024). Additionally, relying on model-generated positive samples can reinforce misleading correlations that stem from flawed intermediate steps, thereby weakening generalization (Setlur et al., 2024). To address these challenges,

recent research has introduced step-level DPO (Setlur et al., 2024; Lai et al., 2024), which offers more granular error identification and thus improves reasoning accuracy. *However, the naive DPO algorithm struggles to capture fine-grained, step-level supervisory signals in multi-step preference learning. This uniform treatment of all correct and incorrect steps significantly limits the model's potential for optimization.*

