# OpenReview forum: "LearNAT: Learning NL2SQL with AST-guided Task Decomposition  for Large Language Models"
_ICLR.cc/2026/Conference — ICLR 2026 Poster_

### Official Review · Reviewer_HybD · 2025-10-27

**Soundness:** 4
**Presentation:** 4
**Contribution:** 3
**Rating:** 8
**Confidence:** 4

**Summary:**

This paper presents LearNAT, a framework designed to enhance the performance of small, open large language models (LLMs) on natural language to SQL (NL2SQL) tasks. The method combines two ideas:

1. An AST-guided decomposition synthesis process that uses Monte Carlo Tree Search (MCTS) guided by the SQL Abstract Syntax Tree (AST) of gold queries to generate verifiable intermediate subtasks (sub-SQLs).

2. A Margin-Aware Direct Preference Optimization (MDPO) objective that introduces AST-based structural margins between positive and negative steps, providing fine-grained reward signals without a learned reward model.

Experiments on BIRD and Spider benchmarks demonstrate substantial accuracy improvements for open Qwen2.5-coder models (7B/14B/32B) and notable efficiency advantages over GPT-4-based system-level pipelines. The paper provides ablations, cost analyses, and code release, emphasizing openness and reproducibility.

**Strengths:**

- **Motivated practical problem:** Tackles a highly relevant challenge — enabling small, public models to achieve competitive NL2SQL performance without expensive test-time pipelines.

- **Strong methodological alignment:** AST-guided decomposition is both interpretable and efficient, providing verifiable supervision that directly matches SQL’s structural nature.

- **Novel preference learning variant:** The margin-aware DPO objective elegantly integrates structured information into preference learning without requiring a learned reward model.

- **Empirical results and cost analysis:** Large gains on BIRD and Spider, along with token-cost comparisons, demonstrate both effectiveness and efficiency.

- **Reproducibility:** Clear method description, ablation studies, and commitment to open code release.

**Weaknesses:**

### Offline Dependence on Gold SQL ASTs
The synthesis process relies on gold ASTs ($AT(Y)$) for search and reward computation, limiting scalability to unlabeled settings.

### Baseline Comparison Fairness
Model-level and system-level results (e.g., GPT-4 pipelines) are mixed without clear labels or cost normalization.

### Limited Reward-Learning Baselines
MDPO is compared only to vanilla DPO.

### Compute and Cost Transparency
Synthesis and fine-tuning costs are underreported.

### Robustness and Variance Reporting
Main results appear from single runs, which limits reliability.

### Scope Limitation to Canonical ASTs
The method depends on well-defined SQL ASTs, which may not exist in less-structured domains.

### Relation to Concurrent Structured-Reasoning Work
The paper should cite Struct-LLM (Stoisser et al., 2025), which also explores structured reasoning over SQL and Cypher using reinforcement learning. Briefly contrast LearNAT’s offline AST-guided preference learning with Struct-LLM’s online RL-based reasoning approach.

**Questions:**

### Method Clarity & Assumptions

1. **Gold AST availability**
   You mention that the decomposition synthesis uses the gold SQL AST to guide MCTS.
   - How does this affect scalability to datasets without gold SQLs?
   - Can LearNAT generate training data in a semi-supervised setting, or does it strictly rely on gold supervision?

2. **Verification signal granularity**
   You mention “verifiable intermediate subtasks.”
   - Are these subtasks verified purely syntactically (AST match) or also semantically (execution match on DB)?
   - How do you handle equivalent but syntactically different SQL forms?

3. **MDPO stability**
   - Did you observe training instability compared to vanilla DPO due to margin scaling or structural rewards?
   - Are the AST-based margins dynamically computed or fixed?

---

> ### Author Response · Authors · 2025-11-22
> **Response to Reviewer HybD (Part I)**
>
> Thank you for your acknowledgment on the motivated practical problem, strong methodological alignment, novel preference learning variant, empirical results and cost analysis and reproducibility. We deeply value the constructive criticisms you raised and respond to each concern in detail below.
>
> ## Response to Offline Dependence on Gold SQL ASTs & Scope Limitation to Canonical ASTs (W1, W6 & Q1)
>
> We sincerely appreciate you for raising this important question, and we are glad to further discuss the applicability of LearNAT in scenarios where GT SQL is unavailable.
>
> 1. **Generalization to Scenarios witout GT SQL:**
>
> Although LearNAT’s data synthesis relies on datasets that contain GT SQL, it is noteworthy that LearNAT demonstrates **strong out-of-distribution (OOD) generalization**. As shown in Table 3 of manuscript, LearNAT achieves a 9.4% improvement on the OOD Spider dataset. Therefore, when facing a dataset without GT SQL, one can still rely on a GT-labeled dataset to train the model, which then generalizes effectively to datasets lacking GT SQL.
>
> 2. **Applicability to Semi-Supervised Settings:**
>
> We believe LearNAT is also suitable for semi-supervised scenarios. As a concrete example, we assume the following semi-supervised scenario: we have a labeled dataset BIRD-train-part1 and an unlabeled dataset BIRD-train-part2, each comprising half of the full BIRD-train dataset. We perform the following semi-supervised learning procedure:
>
> > 1. We use LearNAT to construct task-decomposition data on BIRD-train-part1, where the correctness of each subtask is **verified using the ground-truth AST**.
> > 2. Using the synthesized data from Step 1, we fine-tune the Qwen2.5-Coder-7B model (denoted as LearNAT$^{1}$).
> > 3. We then apply the fine-tuned model from Step 2 to construct task-decomposition data on BIRD-train-part2, **without verifying subtask correctness using a ground-truth AST**.
> > 4. We combine the synthetic data from Steps 1 and 3, and use this merged dataset to fine-tune the original Qwen2.5-Coder-7B model, yielding LearNAT$^{2}$.
>
> The performance of these models is shown in the following table:
>
> | Methods | BIRD |  |  |  | Spider |  |  |  |  |
> |---|---|---|---|---|---|---|---|---|---|
> |  | Simple | Moderate | Challenging | Total | Easy | Medium | Hard | Extra Hard | Total |
> | Qwen2.5-Coder-7B | 56.1 | 34.5 | 33.8 | 47.5 | 82.7 | 84.1 | 71.8 | 54.8 | 77.0 |
> | LearNAT$^{1}$ | 61.4 | 42.2 | 38.9 | 53.5 | 90.3 | 88.8 | 73.6 | 62.0 | 82.3 |
> | LearNAT$^{2}$ | 62.1 | 45.2 | 38.9 | 54.8 | 92.3 | 89.9 | 74.1 | 62.0 | 83.4 |
>
>
> The above experiment demonstrates **the effectiveness of LearNAT in a semi-supervised setting**. LearNAT can rely entirely on a small labeled dataset with ground-truth SQL to initialize the model, and then use the capabilities acquired from this small dataset to annotate the unlabeled portion, thereby reducing dependence on labeled data.
> However, it is important to note that LearNAT$^{2}$ achieves only 54.8% on the BIRD dataset, whereas LearNAT trained with the fully labeled dataset reaches 58.1%. Thus, although LearNAT proves its feasibility in semi-supervised scenarios, the supervised setting remains the more suitable and effective regime for LearNAT.

---

> ### Author Response · Authors · 2025-11-22
> **Response to Reviewer HybD (Part II)**
>
> ## Response to Baseline Comparison Fairness (W2)
>
> We sincerely appreciate your suggestion and agree that this was a substantial omission in the original manuscript. In response to your comment, we follow SuperSQL [1] and report the *Avg. Tokens (k) / Query* and *EX / Avg. Tokens (k)* metrics for these baseline methods. The experimental results are presented below (notably, CatSQL and SENSE are excluded here because neither their code nor checkpoints are publicly available).
>
>  Methods | LLMs | BIRD | Spider | Total | Avg. Tokens (k) / Query | EX / Avg. Tokens (k)  |
> |---|---|---|---|---|---|---|
> | System-Level |  |  |  |  |  |  |
> | C3-SQL | GPT-4 | 50.2 | 82.0 | 63.0 | 21.2 | 3.0 |
> | DIN-SQL | GPT-4 | 50.7 | 74.2 | 60.2 | 9.7 | 6.2 |
> | MetaSQL | GPT-4 | 47.6 | 69.6 | 56.5 | 10.1 | 5.6 |
> | MAG-SQL | GPT-4 | 57.6 | 85.3 | 68.8 | 11.4 | 6.0 |
> | SuperSQL | GPT-4 | 58.5 | 87.0 | 70.0 | 9.1 | 7.7 |
> | MAC-SQL | GPT-4 | 59.4 | 86.7 | 70.4 | 6.5 | 10.8 |
> | Model-Level |  |  |  |  |  |  |
> | ACT-SQL | GPT-4 | 52.4 | 74.5 | 61.3 | 1.6 | 38.3 |
> | DAIL-SQL | GPT-4 | 54.3 | 76.2 | 63.1 | 1.3 | 48.6 |
> | CodeS | CodeS-7B | 57.0 | 85.4 | 68.4 | **1.1** | 62.2 |
> |  | CodeS-15B | 58.5 | 84.9 | 69.1 | **1.1** | **62.8** |
> | Ours |  |  |  |  |  |  |
> | LearNAT | Qwen2.5-Coder-7B | 58.1 | 86.4 | 69.5 | 1.8 | 38.6 |
> |  | Qwen2.5-Coder-14B | 61.2 | 86.9 | 71.6 | 1.7 | 42.1 |
> |  | Qwen2.5-Coder-32B | **65.0** | **88.4** | **74.4** | 1.8 | 41.3 |
>
> The experimental results show that, compared with system-level methods, **LearNAT achieves substantial advantages in terms of performance, token consumption, and performance per token**. When compared with model-level methods (for example, CodeS), although LearNAT attains **higher performance**, it consumes **more tokens**. This is because LearNAT output both the intermediate subtasks and their corresponding SQL statements, which leads to increased token usage.
>
> ## Response to Limited Reward-Learning Baselines (W3)
>
> We sincerely appreciate your suggestion and fully agree with your perspective that introducing additional DPO variants can better demonstrate the effectiveness of MDPO. In response to your comment, we include two highly recognized DPO variants, IPO [2] and KTO [3]. The experimental results are shown in the table below. The results indicate that **MDPO also outperforms these two DPO variants**.
>
> | Methods | BIRD-dev  |  |  |  | Spider-dev |  |  |  |  |
> |---|---|---|---|---|---|---|---|---|---|
> |  | Simple | Moderate | Challenging | Total | Easy | Medium | Hard | Extra Hard | Total |
> | Qwen2.5-Coder-7B |  |  |  |  |  |  |  |  |  |
> | LearNAT | **65.4** | **48.4** | **42.4** | **58.1** | **95.2** | **92.4** | **76.4** | **67.5** | **86.4** |
> | MDPO->KTO | 63.1 | 43.9 | 34.7 | 54.6 | 89.1 | 90.6 | 68.4 | 63.9 | 82.2 |
> | MDPO->IPO | 62.5 | 42.6 | 32.6 | 53.7 | 86.3 | 91.0 | 65.5 | 64.5 | 81.3 |
>
> ## Response to Compute and Cost Transparency (W4)
>
> Thank you very much for your suggestion. In the original manuscript, we have already reported the token cost associated with data synthesis, as shown in Table 1. During fine-tuning, the model is trained for 1 hour × 8 × A100 GPUs, processing 6.9M tokens in the *Warm-up Strategy for Foundational Skill Acquisition* stage, and for 3.5 hours × 8 × A100 GPUs, processing 20.1M tokens in the *DPO with AST Margin* stage.
>
> If there are any other cost-related metrics you are interested in, please let us know, and we will promptly provide the relevant information.
>
> ## Response to Robustness and Variance Reporting (W5)
>
> Thank you for your suggestion. In the table below, we provide the mean and standard deviation of three versions of the LearNAT model over five experimental runs, to ensure the **reliability and robustness** of LearNAT’s performance.
>
> | LLMs | BIRD-dev |  |  |  | Spider-dev |  |  |  |  |
> |---|---|---|---|---|---|---|---|---|---|
> |  | Simple | Moderate | Challenging | Total | Easy | Medium | Hard | Extra Hard | Total |
> | Qwen2.5-Coder-7B | 65.1±1.2 | 47.9±1.4 | 41.4±3.1 | 57.6±1.7 | 94.5±2.1 | 92.6±0.9 | 76.6±2.2 | 66.6±1.0 | 86.2±1.4 |
> | Qwen2.5-Coder-14B | 68.3±0.8 | 51.4±0.9 | 44.6±1.0 | 61.0±1.3 | 95.8±2.4 | 91.6±1.2 | 80.6±3.7 | 68.0±1.4 | 86.9±1.3 |
> | Qwen2.5-Coder-32B | 70.6±0.7 | 55.4±1.0 | 58.3±1.4 | 64.8±1.3 | 96.0±2.1 | 92.5±1.0 | 84.6±2.2 | 69.3±1.5 | 88.3±1.2 |

---

> ### Author Response · Authors · 2025-11-22
> **Response to Reviewer HybD (Part III)**
>
> ## Response to Relation to Concurrent Structured-Reasoning Work (W7)
>
> We sincerely appreciate your pointing us to such excellent prior work. Our comparison between LearNAT and Struct-LLM [4] is as follows:
>
> 1. **Training strategy**. Struct-LLM is trained using online GRPO, whereas LearNAT adopts offline DPO. In terms of time cost, DPO relies on offline training data and therefore does not require rollouts for data sampling during training, which substantially reduces the computational overhead. Regarding memory consumption, the online rollouts in GRPO lead to larger KV caches, resulting in significantly higher GPU memory usage.
>
> 2. **Reward design**. Struct-LLM employs three reward types: LLM Judge Reward, String Matching Reward, and Structural Consistency Reward. The latter two are conceptually similar to the node-level and structure-level AST similarity rewards proposed in LearNAT. The key difference is that Struct-LLM models these rewards at the string level, whereas LearNAT performs reward modeling directly on AST structures.
>
> ## Response to Syntax Verification versus Semantic Verification (Q2.1)
>
> Thank you for raising this question. In LearNAT, we verify subtasks **solely based on syntax**. This is because the correctness of a generated subtask can be **confirmed with reliability** by examining the structural relationships in its AST.
>
> Semantic verification, however, is much more challenging. The **execution result of a subtask** on the database is often **difficult to judge as correct**. For example, if a query requests the average score of students in a class, the execution result of GT SQL returns only the final average score. A subtask might involve counting the number of students in that class, but the result of this subtask is not reflected in the execution result of the GT SQL, making semantic verification infeasible in this case.
>
> ## Response to the Analysis of SQL Diversity (Q2.2)
>
> We sincerely appreciate your constructive comments. Indeed, **a single query may correspond to multiple SQL queries with different structures**. However, because we only have access to a single GT SQL, the MCTS process in the decomposition data synthesis stage may discover alternative valid subtasks that correspond to other correct SQLs. These potential subtasks, despite being semantically valid, are **misclassified as incorrect** because they do not align with the subtask structure derived from the available GT SQL. We **acknowledge** that this reflects a limitation of LearNAT: it lacks sufficient diversity in the synthesized data.
>
> To address this, we designed the following experiment: specifically, we use GLM-4-Plus to rewrite the SQL for each query in the training data. The rewriting is constrained such that the **AST structures** of the original and rewritten SQL are **different**, but their **execution results** are **identical**, thereby enhancing the diversity of the synthesized data. We denote this method as LearNAT+. Our experimental results, as shown below, indicate that increasing data diversity in this way can further improve the performance of LearNAT.
>
> | Methods | BIRD-dev |  |  |  | Spider |  |  |  |  |
> |---|---|---|---|---|---|---|---|---|---|
> |  | Simple | Moderate | Challenging | Total | Easy | Medium | Hard | Extra Hard | Total |
> | Qwen2.5-Coder-7B |  |  |  |  |  |  |  |  |  |
> | LearNAT | 65.4 | 48.4 | 42.4 | 58.1 | 95.2 | 92.4 | 76.4 | 67.5 | 86.4 |
> | LearNAT+ | 66.6 | 49.5 | 46.5 | 59.5 | 96.8 | 93.3 | 83.3 | 68.7 | 88.5 |
>
> We acknowledge that the **strict** AST-based evaluation in LearNAT does indeed reduce the diversity of the model’s outputs. However, while enhancing the diversity of model outputs is certainly a desirable goal, we believe it is not the most critical one. The **primary motivation** behind LearNAT is to teach the model to reason **correctly** subtask by subtask. The strict AST-based supervision guarantees that the synthesized decompositions form a **valid sequence of subtasks**. This guarantee of correctness is the key driver of LearNAT’s performance gains and represents the core contribution of our work.

---

> ### Author Response · Authors · 2025-11-22
> **Response to Reviewer HybD (Part IV)**
>
> ## Response to Training Instability of MDPO (Q3.1)
>
> We appreciate your valuable suggestions. In response to this comment, we have revised our manuscript and added Appendix F.6, which provides an analysis comparing MDPO with DPO. In this subsection, we present the training loss curves of both DPO and MDPO. Based on this comparison, we **did not observe any training instability in MDPO**.
>
> Additionally, based on the comparison of training losses, we observe that **MDPO exhibits a faster and more pronounced training loss reduction compared with DPO**. This observation indicates that the reward margin introduced in MDPO provides larger and more meaningful gradients for model optimization, which can be attributed to MDPO’s ability to exploit the reward differences among negative samples.
>
> ## Response to the computation of AST-based margins (Q3.2)
>
> Since our optimization is performed using DPO, an offline RL algorithm, the training data remain fixed. Consequently, once the collection of paired preference data is completed, the AST-based margins for these pairs are computed and remain **fixed** thereafter.
>
> ## Manuscript Revision Explanation
>
> We would like to once again express our gratitude for your constructive suggestions and acknowledge that the original manuscript has several shortcomings. In response to your feedback, we have revised our manuscript as follows:
>
> 1. **More comparisons with preference-optimization algorithms**: We updated Table 3 in the revised manuscript by adding KTO and IPO as additional ablation baselines to further demonstrate the effectiveness of MDPO.
>
> 2. **Synthesis Diversity**: In Appendix.F.6 of the revised manuscript, we discuss the limitations of LearNAT regarding the diversity of synthesized data and outline potential directions for improvement.
>
> 3. **Semi-supervised setting**: In Appendix.F.7 of the revised manuscript, we provide a detailed discussion of the applicability of LearNAT in semi-supervised scenarios.
>
> 4. **Token consumption comparison**: In Appendix.F.8 of the revised manuscript, we present a comprehensive comparison of token consumption between baseline methods and LearNAT.
>
> 5. **Robustness analysis**: In Appendix.F.9 of the revised manuscript, we conduct multiple experiments and analyze the robustness of LearNAT.
>
> 6. **Margin-aware DPO Analysis**: In Appendix.F.11 of revised manuscript, we  plot the training loss curves of both DPO and MDPO to further demonstrate the effectiveness of MDPO.
>
> 7. **Discussion with Struct-LLM**: In Appendix.G.1 of revised manuscript, we provide a detailed discussion of the differences between LearNAT and Struct-LLM in terms of reward design and how these rewards are applied.
>
> To facilitate quick identification of our modifications, we have highlighted the changes using **special colored fonts**. (Please refer to the *Guideline for Reviewers* on the last page of the revised manuscript.)
>
> ### References:
>
> [1] The Dawn of Natural Language to SQL: Are We Fully Ready? VLDB’24
>
> [2] A General Theoretical Paradigm to Understand Learning from Human Preferences, AISTATS'24
>
> [3] KTO: Model alignment as prospect theoretic optimization, Arxiv'24
>
> [4] STRuCT-LLM: Unifying Tabular and Graph Reasoning with Reinforcement Learning for Semantic Parsing, Arxiv'25

---

### Official Review · Reviewer_RUNy · 2025-10-28

**Soundness:** 3
**Presentation:** 3
**Contribution:** 3
**Rating:** 6
**Confidence:** 4

**Summary:**

The paper propose LearNAT, a framework that enhances LLMs' ability to decompose complex queries through decomposition synthesis procedure and margin-aware reinforcement learning.  Decomposition synthesis procedure uses AST-guided search and pruning to precede efficient and verifiable decomposition on the BIRD-train dataset for the margin-aware reinforcement learning. Margin-aware reinforcement learning modified DPO's loss function by a AST-based reward distinction between samples. The experiment shows that LearNAT enables 7B-parameter models to reach performance close to GPT-4.

**Strengths:**

1. The paper introduces a novel approach that leverages ASTs for task decomposition, enabling the synthesis of training data for reinforcement learning.
2. This paper further proposes a modification to the DPO framework by incorporating an AST-distance-based reward to better estimate reward margins and enhance performance on BIRD and Spider datasets.

**Weaknesses:**

1. The authors acknowledge that although LearNAT does not achieve state-of-the-art performance among system-level approaches, it consumes fewer tokens during inference. However, LearNAT should also be compared against model-level approaches of similar model size, such as Reasoning-SQL and OmniSQL, which demonstrate stronger performance on the BIRD leaderboard.

**Questions:**

None.

---

> ### Author Response · Authors · 2025-11-22
> **Response to Reviewer RUNy (Part I)**
>
> Thank you for your recognition of our AST-guided task decomposition, which highlights its strong controllability and excellent performance in intermediate process validation. We also appreciate your acknowledgment of our innovative enhancement to the DPO algorithm, enabling fine-grained supervision for multi-step reasoning. Your positive feedback is highly encouraging. Below, we address your comments in detail.
>
> ## Response to Additional Baseline Comparisons (W1)
>
> We sincerely thank you for pointing out these newer models, and we are very glad to further discuss the comparison between LearNAT and these SOTA methods [1,2].
>
> We fully **acknowledge** your initial observation that Reasoning-SQL [1] and OmniSQL [2] report higher results than LearNAT on the BIRD dataset in their manuscripts. However, we have noticed a potential issue: **LearNAT and these methods [1,2] employ different evaluation protocols**. It is important to note that LearNAT follows a **widely recognized evaluation protocol**, namely DAIL-SQL [3], whereas the baselines you mentioned do not adhere to this protocol. To enable a fairer comparison, **we use LearNAT’s checkpoints and evaluate its performance under the inference protocols employed by these baseline methods [1,2]**.
>
> Since Reasoning-SQL has not released any checkpoints or code, we do not consider a direct comparison between LearNAT and Reasoning-SQL at this stage. To further complement the evaluation, we incorporate three additional baselines [4,5,6] suggested by Reviewer kPT4, achieving a more diverse set of comparisons.
>
> The experimental results are presented in the table below. The results indicate that LearNAT fully surpasses Syn COT [4], SQL-O1 [5], and Alpha-SQL [6], and although it performs slightly lower than OmniSQL [2] on the BIRD dataset alone, it outperforms OmniSQL overall across the BIRD and Spider datasets.
>
> | Methods | Evaluation Protocol | LLMs | BIRD |  |  |  | Spider |  |  |  |  | Total  |
> |---|---|---|---|---|---|---|---|---|---|---|---|---|
> |  |  |  | Simple | Moderate | Challenging | Total | Easy | Medium | Hard | Extra Hard | Total | |
> | Syn CoT | Syn CoT | Qwen2.5-7B-Instruct |  |  |  | 59.2 |  |  |  |  | 78.9 | 67.1 |
> | LearNAT | Syn CoT | Qwen2.5-Coder-7B | 67.6 | 48.0 | 45.8 | **59.6** | 91.9 | 91.0 | 71.3 | 63.9 | 83.6 | **69.2** |
> | SQL-o1 | SQL-o1 | Llama3-8B | 71.8 | 52.3 | 45.2 | 63.4 | 94.4 | 93.0 | 81.0 | 68.7 | 87.4 | 73.1 |
> | LearNAT | SQL-o1 | Qwen2.5-Coder-7B | **72.5** | **54.2** | **49.3** | **64.8** | **96.4** | **94.8** | **78.2** | **74.1** | **89.1** | **74.6** |
> | Alpha-SQL | Alpha-SQL | Qwen2.5-Coder-7B | 72.6 | 59.3 | 53.1 | 66.8 | 94.0 | 89.2 | 76.4 | 63.3 | 84.0 | 73.7 |
> | LearNAT | Alpha-SQL | Qwen2.5-Coder-7B | **74.4** | **61.5** | **52.8** | **68.4** | **97.2** | **96.0** | **80.5** | **77.1** | **90.6** | **77.4** |
> | OmniSQL | OmniSQL | Qwen2.5-Coder-7B |  |  |  | **63.9** |  |  |  |  | 81.2 | 70.9 |
> | LearNAT | OmniSQL | Qwen2.5-Coder-7B | 68.2 | 50.3 | 50.7 | 61.1 | 96.0 | 91.5 | 77.6 | 65.1 | **86.0** | **71.1** |
>
> **Further Discussion on the Comparison Between LearNAT and OmniSQL:**
>
> Both LearNAT and OmniSQL optimize NL2SQL performance by constructing synthetic datasets. However, their core contributions differ. OmniSQL focuses on synthesizing a **much larger training dataset** with 2.5M queries, whereas LearNAT emphasizes **constructing training data with verifiable intermediate subtasks**, improving data quality without increasing data quantity. Specifically, since LearNAT generates subtask data based on BIRD-Train, the resulting training set contains only 7.2k queries.
>
> We believe that, although OmniSQL achieves higher performance on the BIRD dataset, this **does not conflict with LearNAT’s core contribution**: LearNAT provides a **reliable** mechanism for subtask generation and evaluation. In future work, we plan to further enhance LearNAT’s performance by expanding its training data, for example, by incorporating OmniSQL as a source for subtask decomposition and synthetic data generation.
>
> **Discussion on the Comparison Between LearNAT and Reasoning-SQL:**
>
> Similar to LearNAT, Reasoning-SQL also employs reinforcement learning to optimize LLM performance on NL2SQL and replaces naïve binary rewards with continuous reward signals. However, an **important distinction** lies in where the reward is applied. Although Reasoning-SQL designs five different reward mechanisms, all of these rewards are **applied only to the final generated SQL**.
>
> In contrast, the rewards designed in LearNAT are primarily used to **evaluate intermediate subtasks**. In other words, LearNAT introduces a *process reward* model that explicitly regulates the correctness of intermediate reasoning steps, whereas Reasoning-SQL focuses on *outcome rewards*. Existing literature [7,8] consistently indicates that process rewards, by providing richer and more fine-grained learning signals, can more effectively improve model performance than outcome rewards.

---

> ### Author Response · Authors · 2025-11-22
> **Response to Reviewer RUNy (Part II)**
>
> ## Manuscript Revision Explanation
>
> Thank you again for your insightful suggestions. In response to your feedback, we have revised our manuscript as follows:
>
> 1. **Performance comparison with OmniSQL**: In revised manuscript, we have incorporated more baseline methods, including OmniSQL [2], and provided a thorough comparison, along with a discussion on the demand for training data volume.
>
> 2. **Discussion with Reasoning-SQL**: In Appendix.G.1 of revised manuscript, we provide a detailed discussion of the differences between LearNAT and Reasoning-SQL in terms of reward design and how these rewards are applied.
>
> To facilitate quick identification of our modifications, we have highlighted the changes using **special colored fonts**. (Please refer to the *Guideline for Reviewers* on the last page of the revised manuscript.)
>
> ### References:
>
> [1] Reasoning-SQL: Reinforcement Learning with SQL Tailored Partial Rewards for Reasoning-Enhanced Text-to-SQL, Arxiv'25
>
> [2] Omnisql: Synthesizing high-quality text-to-sql data at scale, Arxiv'25
>
> [3] Text-to-sql empowered by large language models: A benchmark evaluation, VLDB'24
>
> [4] Uncovering the Impact of Chain-of-Thought Reasoning for Direct Preference Optimization: Lessons from Text-to-SQL, ACL'25
>
> [5] SQL-o1: A Self-Reward Heuristic Dynamic Search Method for Text-to-SQL, Arxiv'25
>
> [6] Alpha-SQL: Zero-Shot Text-to-SQL using Monte Carlo Tree Search, ICML'25
>
> [7] Let’s verify step by step, ICLR'24
>
> [8] Math-Shepherd: Verify and Reinforce LLMs Step-by-step without Human Annotations, ACL'24

---

> > ### Comment · Reviewer_RUNy · 2025-11-27
> >
> > Thank you for your response to additional baseline comparisons. Your experimental results for LearNAT, evaluated under different methods and corresponding protocols, clearly demonstrate its strong performance. I have raised my score to 8 to further endorse this paper.

---

> > > ### Author Response · Authors · 2025-11-27
> > > **Thanks for acknowledging our responses**
> > >
> > > Dear Reviewer RUNy,
> > >
> > > Thank you for your time, effort, and positive evaluation. Your guidance and insights are essential for improving the quality of our work.
> > >
> > > We once again appreciate your thoughtful feedback and guidance.
> > >
> > > Best wishes,
> > >
> > > All authors of submission 22830

---

### Official Review · Reviewer_kPT4 · 2025-11-06

**Soundness:** 3
**Presentation:** 2
**Contribution:** 2
**Rating:** 4
**Confidence:** 5

**Summary:**

This paper introduces LearNAT, a novel framework that starts from task decomposition for NL2SQL. The core innovation lies in leveraging an AST-guided Monte Carlo Tree Search (MCTS) reasoning framework for efficient reasoning and data synthesis, as well as integrating AST-based structural alignment into the optimization objective to enhance the DPO algorithm. These methods collectively boost the baseline model’s performance by more than 10%.

**Strengths:**

1. Proposes an AST-guided Chain-of-Thought (CoT) task decomposition and verification mechanism, achieving high controllability and impressive success rates in intermediate process validation.
2. Innovatively improves the DPO algorithm by incorporating AST skeleton contrast in the optimization target, enabling fine-grained supervision of multi-step reasoning.

**Weaknesses:**

1. The writing lacks clarity, particularly regarding the model inference stage: implementation details, methods used, and specific parameters are not sufficiently described. It remains unclear whether Monte Carlo Tree Search (MCTS) or voting methods were employed during the inference process. Furthermore, the rationale behind the specific parameter settings is not discussed, nor is it specified whether hyperparameter analysis was conducted to optimize the inference performance.
2. The baseline selection in this paper is notably insufficient and lacks relevance. Current comparisons fail to directly target key methods such as DPO [1] and MCTS [2,3] with similar model scales, making LearNAT's claimed advantages difficult to substantiate. Without rigorous and fair evaluations against established approaches, the performance improvements may be unconvincing. The authors must provide more targeted and transparent baseline comparisons to truly demonstrate the superiority of LearNAT.
3. Although the abstract and introduction highlight the heavy test-time computational burden of existing methods, the paper does not explicitly quantify the inference efficiency gains brought by AST-pruned MCTS, nor provide detailed time cost statistics. Supplementary experiments in this regard are recommended.


[1] Uncovering the Impact of Chain-of-Thought Reasoning for Direct Preference Optimization: Lessons from Text-to-SQL

[2] SQL-o1: A Self-Reward Heuristic Dynamic Search Method for Text-to-SQL

[3] Alpha-SQL: Zero-Shot Text-to-SQL using Monte Carlo Tree Search

**Questions:**

1. Given the diversity of SQL queries—where different SQL skeletons result in varying AST structures—how does the AST-guided MCTS handle such cases during data synthesis? Are these instances treated as error trajectories?
2. Can the authors provide a more detailed analysis of sample correctness to elucidate the intrinsic incentives of the improved DPO? Specifically, it would be helpful to demonstrate under what kinds of samples LearNAT’s margin-aware DPO exhibits advantages over vanilla DPO, rather than only presenting final aggregate metrics.

---

> ### Author Response · Authors · 2025-11-22
> **Response to Reviewer kPT4 (Part I)**
>
> Thank you for your acknowledgment on the AST-guided task decomposition, the verification mechanism and our innovative enhancement to the DPO algorithm. We are especially grateful for your critical comments, which prompted deep reflection on the limitations of our original submission and motivated us to substantially improve our work.
>
> ## Clarification of Potential Misunderstanding
>
> We sincerely apologize for the potential misunderstanding caused by the insufficient description of the inference stage in our manuscript. Here, we provide a detailed clarification:
>
> > During inference, we strictly follow the evaluation protocol provided by DAIL-SQL [1] (the *Without Voting setting*). The protocol provides a complete set of prompts to better structure the instructions, user queries, and database schema information, enabling the LLM to generate a **single response** from which the SQL statement is extracted as the final answer.
>
> In other words, **LearNAT does not employ any search or voting strategy during inference**. MCTS is used **only** in the decomposition data synthesis stage, **not** during inference.
> It is worth noting that in Appendix.Fig.9 of the original manuscript, we used multi-sampling and self-consistency strategies to improve the performance of LearNAT. However, the purpose of this section is to demonstrate that LearNAT possesses the **potential** to **further improve performance** by employing search and voting strategies during inference. In the core experimental results of this manuscript, such as the baseline comparisons in Table 2 and the ablation studies in Table 3, LearNAT was evaluated **without incorporating any search or voting strategies**.
>
> ## Response to Hyperparameter Analysis in the Inference Stage (W1)
>
> We sincerely appreciate your suggestion and fully understand the importance of hyperparameter sensitivity analysis. As noted above, LearNAT does not employ any parameter optimization techniques during inference and therefore introduces no additional hyperparameters. If you are interested in analyzing any existing inference-related hyperparameters (e.g., top-k, temperature), please let us know, we will **promptly** conduct the analysis and **incorporate the results into the revised manuscript**.
>
> ## Response to Additional Baseline Comparisons (W2)
>
> We sincerely appreciate the baselines you suggested and fully agree that conducting comprehensive comparisons with the latest methods is essential for rigorously evaluating the superiority of LearNAT.
>
> Before presenting the comparisons, we would like to clarify the following points: as noted earlier, LearNAT follows the evaluation protocol established by prior work, DAIL-SQL [1], which is a **widely recognized and authoritative benchmark**. However, the methods you listed [2,3,4] **do not follow** this evaluation protocol. For example, SQL-O1 [3] and Alpha-SQL [4] both introduce MCTS during inference to generate multiple candidate SQL statements. Since these works [2,3,4] have not released their model checkpoints, and to ensure a fairer comparison, we instead **employ LearNAT’s checkpoints and evaluate its performance under the evaluation protocols used in those methods [2,3,4]**. The experimental results are presented in the table below.
>
> | Methods | Evaluation Protocol | LLMs | BIRD |  |  |  | Spider |  |  |  |  |
> |---|---|---|---|---|---|---|---|---|---|---|---|
> |  |  |  | Simple | Moderate | Challenging | Total | Easy | Medium | Hard | Extra Hard | Total |
> | Syn CoT [2] | Syn CoT | Qwen2.5-7B-Instruct |  |  |  | 59.2 |  |  |  |  | 78.9 |
> | LearNAT | Syn CoT | Qwen2.5-Coder-7B | 67.6 | 48.0 | 45.8 | **59.6** | 91.9 | 91.0 | 71.3 | 63.9 | **83.6** |
> | SQL-o1 [3] | SQL-o1 | Llama3-8B | 71.8 | 52.3 | 45.2 | 63.4 | 94.4 | 93.0 | 81.0 | 68.7 | 87.4 |
> | LearNAT | SQL-o1 | Qwen2.5-Coder-7B | **72.5** | **54.2** | **49.3** | **64.8** | **96.4** | **94.8** | **78.2** | **74.1** | **89.1** |
> | Alpha-SQL [4] | Alpha-SQL | Qwen2.5-Coder-7B | 72.6 | 59.3 | 53.1 | 66.8 | 94.0 | 89.2 | 76.4 | 63.3 | 84.0 |
> | LearNAT | Alpha-SQL | Qwen2.5-Coder-7B | **74.4** | **61.5** | **52.8** | **68.4** | **97.2** | **96.0** | **80.5** | **77.1** | **90.6** |
>
> The experimental results indicate that, **under the same evaluation protocol, LearNAT surpasses all existing baseline methods**. Notably, the overall performance of LearNAT can be further improved if additional search is introduced during inference. For example, when applying the inference-time search strategy of Alpha-SQL, LearNAT’s performance on the BIRD dataset increases from 58.1% to 68.4%. However, we also note a significant concern: the token consumption during inference rises sharply, from 1.8k tokens per query to 204.5k tokens per query.

---

> ### Author Response · Authors · 2025-11-22
> **Response to Reviewer kPT4 (Part II)**
>
> ## Response to Token Savings of AST-pruned MCTS and Reduction in Inference Time of LearNAT (W3)
>
> We sincerely appreciate your valuable suggestion and fully agree that discussing token consumption is crucial.
>
> In LearNAT, AST-pruned MCTS is applied **only** during the data synthesis stage and is **not** used during inference. In Table 1 of the original manuscript, we have already reported the benefits of AST-pruned MCTS in reducing token usage, demonstrating a **60.00% reduction in token consumption**.
>
> What we refer to as **high inference-time overhead** specifically concerns methods that introduce multiple sampling steps, MCTS-based search (e.g., SQL-o1 and Alpha-SQL as you mentioned), or multi-round iterative SQL refinement during inference. These procedures substantially increase computational cost at inference time.
>
> In contrast, **LearNAT does not introduce any search or voting strategies during the inference stage**. As an end-to-end approach that directly outputs the final SQL, it achieves performance that is either superior to or competitive with those methods that rely on such additional inference-time strategies.
> For clarity, we summarize the inference-time overhead of these methods in the table below (with LearNAT normalized to 1). The experimental results demonstrate that, while achieving comparable performance, **LearNAT achieves a substantially lower inference-time cost**.
>
> | Methods | Inference Time |
> |---|---|
> | C3-SQL | 11.8 |
> | DIN-SQL | 5.4 |
> | MetaSQL | 5.6 |
> | MAG-SQL | 6.3 |
> | SuperSQL | 5.1 |
> | MAC-SQL | 3.6 |
> | SQL-o1 | 5.3 |
> | Alpha-SQL | 111.2 |
> | LearNAT | 1.0 |
>
> ## Response to the Analysis of SQL Diversity (Q1)
>
> We sincerely appreciate your constructive comments. As you noted, during the MCTS-based subtask search process, some potentially correct subtasks are indeed generated. However, because the ASTs of these subtasks do not exist as subtrees in the AST of the ground-truth SQL, they are **mistakenly judged as incorrect trajectories**. We **acknowledge** that this reflects a limitation of LearNAT: it lacks sufficient diversity in the synthesized data.
>
> To address this, we designed the following experiment: specifically, we use GLM-4-Plus to rewrite the SQL for each query in the training data. The rewriting is constrained such that the **AST structures** of the original and rewritten SQL are **different**, but their **execution results** are **identical**, thereby enhancing the diversity of the synthesized data. We denote this method as LearNAT+. Our experimental results, as shown below, indicate that increasing data diversity in this way can further improve the performance of LearNAT.
>
> | Methods | BIRD-dev |  |  |  | Spider |  |  |  |  |
> |---|---|---|---|---|---|---|---|---|---|
> |  | Simple | Moderate | Challenging | Total | Easy | Medium | Hard | Extra Hard | Total |
> | Qwen2.5-Coder-7B |  |  |  |  |  |  |  |  |  |
> | LearNAT | 65.4 | 48.4 | 42.4 | 58.1 | 95.2 | 92.4 | 76.4 | 67.5 | 86.4 |
> | LearNAT+ | 66.6 | 49.5 | 46.5 | 59.5 | 96.8 | 93.3 | 83.3 | 68.7 | 88.5 |
>
> We acknowledge that the **strict** AST-based evaluation in LearNAT does indeed reduce the diversity of the model’s outputs. However, while enhancing the diversity of model outputs is certainly a desirable goal, we believe it is not the most critical one. The **primary motivation** behind LearNAT is to teach the model to reason **correctly** subtask by subtask. The strict AST-based supervision guarantees that the synthesized decompositions form a **valid sequence of subtasks**. This guarantee of correctness is the key driver of LearNAT’s performance gains and represents the core contribution of our work.

---

> ### Author Response · Authors · 2025-11-22
> **Response to Reviewer kPT4 (Part III)**
>
> ## Response to Differences between DPO and Margin-aware DPO (Q2)
>
> We greatly appreciate your valuable suggestion. We are pleased to further discuss the insights behind our margin-aware DPO design and believe that clarifying the idea will strengthen the manuscript.
>
> ### 1. Case Analysis
>
> Concretely, consider the following case:
>
> **NL query**
>
> > *Consider the average difference between K-12 enrollment and 15-17 enrollment of schools that are locally funded, list the names and DOC type of schools which has a difference above this average.*
>
> **GT SQL**
>
> ```sql
> SELECT T2.School, T2.DOC
> FROM frpm AS T1 INNER JOIN schools AS T2
> ON T1.CDSCode = T2.CDSCode
> WHERE T2.FundingType = 'Locally funded'
>   AND (T1.`Enrollment (K-12)` - T1.`Enrollment (Ages 5-17)`) > (
>     SELECT AVG(T3.`Enrollment (K-12)` - T3.`Enrollment (Ages 5-17)`)
>     FROM frpm AS T3 INNER JOIN schools AS T4
>     ON T3.CDSCode = T4.CDSCode
>     WHERE T4.FundingType = 'Locally funded'
> )
> ```
>
> In the NL2SQL task decomposition for this example, we obtain the following correct subtask and sub-SQL (denoted Response #1):
>
> **Subtask**:
>
> > *Compute the average difference between K–12 enrollment and 15–17 enrollment for locally funded schools.*
>
> **Sub-SQL**:
> ```sql
> SELECT AVG(T1.`Enrollment (K-12)` - T1.`Enrollment (Ages 5-17)`) AS avg_diff
> FROM frpm AS T1
> INNER JOIN schools AS T2
>   ON T1.CDSCode = T2.CDSCode
> WHERE T2.FundingType = 'Locally funded';
> ```
>
> We also collected two erroneous responses. The first erroneous response (Response #2) is:
>
> **Subtask**:
>
> > *Compute the average difference between K–12 enrollment and 15–17 enrollment.*
>
> **Sub-SQL**:
> ```sql
> SELECT AVG(T1.`Enrollment (K-12)` - T1.`Enrollment (Ages 5-17)`) AS avg_diff
> FROM frpm AS T1
> INNER JOIN schools AS T2
>   ON T1.CDSCode = T2.CDSCode;
> ```
>
> The second erroneous response (Response #3) is:
>
> **Subtask**:
>
> > *Compute the average difference between K–12 enrollment and 15–17 enrollment for locally funded schools.*
>
> **Sub-SQL**:
> ```sql
> SELECT AVG(T1.`Enrollment (K-12)` - T1.`Enrollment (Ages 5-17)`) AS avg_diff
> FROM frpm AS T1
> INNER JOIN schools AS T2
>   ON T1.CDSCode = T2.CDSCode
> WHERE T2.FundingType = 'Locally';
> ```
>
> We observe that Response #2 is a **more severe error** than Response #3: Response #3 only uses an incorrect `FundingType` value (a string-level mistake), whereas Response #2 omits the funding filter entirely and thus computes the average across all funding types, yielding a fundamentally incorrect statistic.
>
> However, in standard DPO optimization, Response #2 and Response #3 are treated as **equivalent** rejected samples, despite exhibiting clearly different degrees of error. We argue that such differences should be explicitly distinguished. Therefore, we introduce Margin-aware DPO, which incorporates an AST-based metric as an **offset** term in the DPO loss to differentiate rejected samples according to their error severity.
>
> Under this offset mechanism, the reward margin between Response #1 and Response #2 becomes larger than that between Response #1 and Response #3. We believe that dynamically adjusting the reward margin between the chosen sample and rejected samples of varying error levels enables the model to better distinguish among incorrect responses, thereby allowing it to perform more fine-grained preference optimization.
>
> ### 2. Training Loss Analysis
>
> In Appendix F.11 of revision manuscript, we present the training loss curves of both DPO and MDPO. Based on the comparison of training losses, we observe that MDPO exhibits a faster and more pronounced training loss reduction compared with DPO. This observation indicates that the reward margin introduced in MDPO provides larger and more meaningful gradients for model optimization, which can be attributed to MDPO’s ability to exploit the reward differences among negative samples.

---

> ### Author Response · Authors · 2025-11-22
> **Response to Reviewer kPT4 (Part IV)**
>
> ## Manuscript Revision Explanation
>
> We would like to once again express our gratitude for your constructive suggestions and acknowledge that the original manuscript has several shortcomings. In response to your feedback, we have revised our manuscript as follows:
>
> 1. **Implementation Details**: We have revised Appendix.F.1 of our manuscript, in which we provide a detailed description of the implementation details of our method. Additionally, we have supplemented the section with the inference-stage implementation strategies to prevent any potential misunderstandings by the readers.
>
> 2. **Additional Baselines**: In revised manuscript, we have incorporated more baseline methods, including those you recommended [2,3,4], and provided a thorough comparison between LearNAT and these methods, along with a discussion on the substantial token consumption incurred when introducing search during inference.
>
> 3. **Comparison of Inference Time**: In Appendix.F.5 of revised manuscript, we have supplemented the inference time of LearNAT and compared it with the baseline to demonstrate the advantage of LearNAT in inference time.
>
> 4. **Synthesis Diversity**: In Appendix.F.6 of revised manuscript, we discuss the limitations of LearNAT in terms of synthesized data diversity and propose possible directions for improvement.
>
> 5. **Margin-aware DPO Analysis**: In Appendix.F.10 and F.11 of revised manuscript, we analyze our insights behind the design of MDPO through a case study and plot the training loss curves of both DPO and MDPO to further demonstrate the effectiveness of MDPO.
>
> To facilitate quick identification of our modifications, we have highlighted the changes using **special colored fonts**. (Please refer to the *Guideline for Reviewers* on the last page of the revised manuscript.)
>
> ### References:
>
> [1] Text-to-sql empowered by large language models: A benchmark evaluation, VLDB'24
>
> [2] Uncovering the Impact of Chain-of-Thought Reasoning for Direct Preference Optimization: Lessons from Text-to-SQL, ACL'25
>
> [3] SQL-o1: A Self-Reward Heuristic Dynamic Search Method for Text-to-SQL, Arxiv'25
>
> [4] Alpha-SQL: Zero-Shot Text-to-SQL using Monte Carlo Tree Search, ICML'25

---

> > ### Comment · Reviewer_kPT4 · 2025-11-23
> >
> > Thank you for your response. It resolved part of my concerns, but based on your explanation I still have the following questions:
> >
> > Could you provide a clearer description of the evaluation protocols used in SQL-o1 and Alpha-SQL, and explain how LearNAT is adapted and transferred to these settings? Clarifying these methodological details is essential for a fair comparison.
> >
> > In addition, the experiments involving baseline comparisons and the use of execution results in MCTS to promote diversity should be presented in the main paper to improve the robustness of the work.

---

> > > ### Author Response · Authors · 2025-11-24
> > > **Response to Reviewer kPT4 (Part I)**
> > >
> > > We sincerely thank you for your patience in reviewing our previous responses and for providing further feedback. We acknowledge the shortcomings in our manuscript and are eager to engage in a deeper discussion to address these issues. We greatly value your insightful suggestions, which are crucial to improving the quality of our work.
> > >
> > >
> > > ## Response to description of the evaluation protocols used in SQL-o1 and Alpha-SQL
> > >
> > >
> > > We sincerely thank you for your valuable feedback. We fully agree that clarifying the evaluation protocols of SQL-o1 [1] and Alpha-SQL [2] is essential for improving the clarity of our manuscript. Below, we sequentially describe the evaluation protocols employed by SQL-o1 and Alpha-SQL.
> > >
> > > Before providing a detailed description of the evaluation protocols for SQL-o1 and Alpha-SQL, we wish to reiterate a **key distinction** between the evaluation protocol employed by LearNAT and those used by SQL-o1 and Alpha-SQL:
> > > > Both SQL-o1 and Alpha-SQL incorporate Monte Carlo Tree Search (MCTS) during inference, which enables systematic exploration and generation of a larger set of candidate SQL queries. This contrasts with LearNAT’s original evaluation setting, which does not involve such search-based expansion at inference time.
> > >
> > > This design aligns with a well-established observation in the literature: when LLMs generate multiple candidate outputs, strategies such as best-of-N sampling [3] or self-consistency voting [4] can substantially improve performance—albeit at the cost of significantly increased token consumption due to the need for generating and processing numerous candidates.
> > >
> > > + **Evaluation protocols of SQL-o1**: SQL-o1 employs MCTS during inference. Specifically, at each node of the MCTS tree, SQL-o1 generates a subtask along with its corresponding SQL statement, a mechanism closely resembling that of LearNAT. The complete task chain relevant to the user query is defined by the trajectory from the root node to a leaf node, based on which the final SQL query is constructed. Furthermore, SQL-o1 defines a reward for each node by evaluating the model’s confidence—i.e., the probability assigned by the LLM to the output at that node—and selects the node with the highest confidence. Nodes whose confidence falls below a predefined threshold are pruned and not further expanded.
> > >
> > > + **Adapt LearnNAT to SQL-o1**: When adapting LearNAT to the SQL-o1 framework, we leverage LearNAT to generate both subtasks and subSQLs at each MCTS node. The tree traversal and pruning decisions are guided by the same reward mechanism used in SQL-o1, and the path with the highest cumulative confidence is ultimately selected as the final SQL query.
> > >
> > > + **Evaluation protocols of Alpha-SQL**: Alpha-SQL also integrates MCTS during inference. However, unlike SQL-o1, Alpha-SQL does not decompose the problem into explicit subtasks; instead, each node directly corresponds to the user query and an associated SQL statement. Consequently, while SQL-o1 defines its action space as the generation of the next planning step and its corresponding SQL, Alpha-SQL adopts a richer set of atomic actions: *Rephrase Question*, *Schema Selection*, *Column Value Identification*, *Column Function Identification*, *SQL Generation*, *SQL Revision*, and *Termination*. Through sequential selection of these actions, Alpha-SQL iteratively refines both the natural language query representation and the corresponding SQL.Alpha-SQL also defines node rewards based on the consistency frequency of paths obtained via high-temperature sampling over action sequences. Upon completion of the MCTS search, Alpha-SQL aggregates all SQL queries generated across trajectories and selects the final output based on self-consistency, i.e., agreement among execution results of candidate SQL queries.
> > >
> > > + **Adapt LearnNAT to Alpha-SQL**: In migrating LearNAT to the Alpha-SQL setting, we adopt a straightforward strategy: every LLM invocation within Alpha-SQL is replaced with LearNAT’s model parameters. Notably, for the *SQL Generation* action, we retain LearNAT’s original prompting scheme to ensure that SQL is still produced in a subtask-by-subtask manner. For all other actions, we follow Alpha-SQL’s original prompting design.
> > >
> > > In response to your suggestion, we have revised `Appendix E: Baseline Solutions` to provide a detailed description of the evaluation protocols for both SQL-o1 and Alpha-SQL. Additionally, we have revised `Appendix F.1: Implementation Details` to explain how LearNAT is integrated into these MCTS-based evaluation protocols.

---

> > > > ### Author Response · Authors · 2025-11-24
> > > > **Response to Reviewer kPT4 (Part II)**
> > > >
> > > > ## Response to manuscript revision
> > > >
> > > > We greatly appreciate your insightful comments. In our previous revision, our primary concern was adhering to the **page limit** for the main manuscript. However, we have confirmed that ICLR permits authors to **expand the main text up to 10 pages during the rebuttal phase**. Accordingly, we will move the comparative experiments involving MCTS into the main body of the manuscript and revise `Section 4.2: Experimental Results` to clearly highlight the advantages of LearNAT under MCTS-based evaluation protocols.
> > > >
> > > > ---
> > > >
> > > > We sincerely thank you for the time and effort you have dedicated to reviewing our manuscript. We believe that, under your insightful guidance, the quality and clarity of our work have been significantly enhanced. We will be sure to incorporate all of your suggestions into our revised manuscript. Thank you once again for your constructive and invaluable comments.
> > > >
> > > > Best regards,
> > > >
> > > > All Authors
> > > >
> > > > ### References
> > > >
> > > > [1] SQL-o1: A Self-Reward Heuristic Dynamic Search Method for Text-to-SQL, Arxiv'25
> > > >
> > > > [2] Alpha-SQL: Zero-Shot Text-to-SQL using Monte Carlo Tree Search, ICML'25
> > > >
> > > > [3] Scaling LLM Test-Time Compute Optimally can be More Effective than Scaling Model Parameters, Arxiv'24
> > > >
> > > > [4] Self-consistency improves chain of thought reasoning in language models. ICLR'23

---

> > > > > ### Comment · Reviewer_kPT4 · 2025-11-25
> > > > >
> > > > > Thank you for your response. I will adjust my score accordingly.

---

> > > > > > ### Author Response · Authors · 2025-11-25
> > > > > > **Thank you for acknowledging our responses**
> > > > > >
> > > > > > Dear Reviewer kPT4,
> > > > > >
> > > > > > Thank you for your careful reading of our manuscript and rebuttal, and for your thoughtful and positive response. Your suggestions are invaluable to us, and we truly appreciate every opportunity to engage in this exchange, which is essential for improving the quality of our work.
> > > > > >
> > > > > > Once again, we are sincerely grateful for your kind guidance.
> > > > > >
> > > > > > Best wishes,
> > > > > >
> > > > > > All authors of submission 22830

---

### Author Response · Authors · 2025-12-03
**General Rebuttal Summary**

Dear Program Chairs, Senior Area Chairs, Area Chairs, and Reviewers,

We would like to express our sincere gratitude to all reviewers for their positive recognition of our work and for their insightful and constructive feedback.

## Key Strengths Highlighted by Reviewers

1. **AST-based Task Decomposition**: **All reviewers** (Reviewer kPT4 S1, Reviewer RUNy S1, Reviewer HybD S2) acknowledged the effectiveness of our AST-based NL2SQL task decomposition, which directly aligns with the structural properties of SQL and provides a verifiable decomposition mechanism.

2. **Margin-aware DPO Design**: **All reviewers** (Reviewer kPT4 S2, Reviewer RUNy S2, Reviewer HybD S3) recognized the merit of our margin-aware, AST-guided DPO paradigm, which enables finer-grained preference alignment.

3. **Additional Key Strengths**: Reviewer HybD further appreciated our motivation to enable small public models to achieve competitive NL2SQL performance without expensive test-time pipelines (S1), acknowledged the strong empirical results (S4), and commended the reproducibility of our manuscript (S5).

It is noteworthy that the two strengths unanimously endorsed **by all reviewers**—*AST-based Task Decomposition* and *Margin-aware DPO Design*—**perfectly match the contributions stated in our manuscript introduction**(line 96 and line 105), further confirming that LearNAT’s intended contributions were explicitly recognized by all reviewers.

## Reviewers’ Concerns We Addressed

We also value the reviewers’ insightful comments and constructive suggestions, which have been instrumental in improving the quality of our work. During the rebuttal period, we provided the following responses and additions:

1. **Clarification of Search-free Inference Mode** (Reviewer kPT4 W1): We clarified the setup of our search-free inference mode to Reviewer kPT4 and revised the manuscript to make this point explicit, **which the reviewer acknowledged**.

2. **Additional Baseline Comparisons** (Reviewer kPT4 W2, Reviewer RUNy W1, Reviewer HybD W7): All reviewers suggested new baselines and requested more comparisons. Following their guidance, we conducted extensive comparisons. Results show that **LearNAT achieves superior performance under fair evaluation settings**, and this conclusion was **unanimously endorsed by Reviewer kPT4 and Reviewer RUNy**. Notably, **all** reviewer suggested baselines are from papers **published in 2025**, and outperforming these most recent methods further validates LearNAT’s advantage.

3. **Quantitative Cost Analysis** (Reviewer kPT4 W3, Reviewer HybD W2 & W4): We evaluated inference time, token cost, training duration, GPU memory usage, and other cost metrics for LearNAT and competing baselines. The analysis, which **highlights LearNAT’s cost efficiency**, was **recognized by Reviewer kPT4**.

4. **SQL Diversity** (Reviewer kPT4 Q1, Reviewer HybD Q2.2): We enhanced SQL diversity to further improve LearNAT’s performance and clarified that SQL diversity, while beneficial, is not the primary factor driving NL2SQL performance gains. **Our experiments and conclusions were accepted by Reviewer kPT4**.

5. **Further Discussion on Margin-aware DPO** (Reviewer kPT4 Q2, Reviewer HybD Q3.1): We provided a case study to illustrate insights behind our margin-aware DPO, included training loss curves to demonstrate stability and advantages over vanilla DPO, and introduced two additional DPO variants as baselines to empirically validate the superiority of our method. **Reviewer kPT4 acknowledged our response**.

6. **Dependence of LearNAT on GT SQL** (Reviewer HybD W1, W6 & Q1): We demonstrated LearNAT’s applicability in settings without GT SQL by evaluating its strong performance on OOD datasets and through simulated experiments validating its effectiveness in semi-supervised scenarios. These results confirm that **LearNAT is not limited to settings with GT SQL**.

7. **Robustness Analysis of LearNAT** (Reviewer HybD W5): We assessed robustness by reporting mean and standard deviation across multiple runs. Results show that **LearNAT delivers reliable and stable performance**.

For the additional experiments above, we have provided further analysis and insights in our detailed responses, and we confirm that **the new results continue to support the primary conclusions of our original manuscript**. We have incorporated **all the above additions into the revised manuscript** to further improve its quality.

Once again, we sincerely thank the Program Chairs, Senior Area Chairs, Area Chairs, and Reviewers for your time and dedication in reviewing our submission.

Best regards,

All authors of submission 22830

---

### Meta-Review · Area_Chair_C2ip · 2026-01-08

**Summary:**

This paper demonstrates the effectiveness of an AST-based task decomposition framework for NL2SQL, which aligns naturally with the structural properties of SQL and enables a verifiable decomposition process. The proposed approach, together with the margin-aware DPO design, represents a novel contribution to the NL2SQL literature, particularly in enabling smaller, publicly available models to achieve competitive performance without relying on expensive test-time pipelines.

Across the reviews, several concerns were raised. Multiple reviewers noted a lack of clarity in the description of the inference procedure. Important implementation details—such as whether Monte Carlo Tree Search, voting, or other inference strategies are employed, how key hyperparameters are selected, and whether sensitivity analyses were conducted—are insufficiently specified. This limits reproducibility and makes it harder to fully attribute the observed performance gains.

Reviewers also expressed concerns about baseline selection and evaluation fairness. The current experimental comparisons do not sufficiently cover strong and relevant baselines of comparable model scale, including both system-level approaches (e.g., MCTS-based methods) and competitive model-level methods such as DPO variants, Reasoning-SQL, and OmniSQL.

Despite these shortcomings, I believe the core idea is sound and meaningful. The AST-based task decomposition provides a principled way to incorporate structural inductive bias into NL2SQL, and the empirical results suggest that this design can substantially reduce inference cost while maintaining strong performance.

I prefer to accept it.

**Reviewer Concerns:**

I believe that most of the reviewers’ concerns have been adequately addressed in the rebuttal. In particular, the authors substantially strengthened the experimental evaluation by adding more targeted and extensive baseline comparisons under fair and comparable settings. The new results demonstrate that LearNAT achieves superior performance relative to relevant baselines.

In addition, the authors provided further analysis on SQL diversity, enhancing the diversity of synthesized SQL to improve overall performance. Importantly, they clarified that while SQL diversity contributes positively, it is not the primary factor driving the observed NL2SQL performance gains.

**Reviewer Scores:**

Reviewer kPT4: This reviewer indicated that their major concerns were addressed in the rebuttal and accordingly increased their score from 4 to 6.

Reviewer RUNy: The additional experimental results demonstrated strong performance, leading this reviewer to raise their score from 6 to 8.

Reviewer HybD: This reviewer did not raise any further questions following the rebuttal and is therefore expected to maintain their score of 8.

---

### Decision · Program_Chairs · 2026-01-26

Accept (Poster)